# Towards Energy Efficient Spiking Neural Networks: An Unstructured Pruning Framework

**Xinyu Shi[1,2], Jianhao Ding[2], Zecheng Hao[2] & Zhaofei Yu[1,2]***

[1] Institute for Artificial Intelligence, Peking University.
[2] School of Computer Science, Peking University
{xyshi,djh01998,haozecheng,yuzf12}@pku.edu.cn

## Abstract

Spiking Neural Networks (SNNs) have emerged as energy-efficient alternatives to Artificial Neural Networks (ANNs) when deployed on neuromorphic chips. While recent studies have demonstrated the impressive performance of deep SNNs on challenging tasks, their energy efficiency advantage has been diminished. Existing methods targeting energy consumption reduction do not fully exploit sparsity, whereas powerful pruning methods can achieve high sparsity but are not directly targeted at energy efficiency, limiting their effectiveness in energy saving. Furthermore, none of these works fully exploit the sparsity of neurons or the potential for unstructured neuron pruning in SNNs. In this paper, we propose a novel pruning framework that combines unstructured weight pruning with unstructured neuron pruning to maximize the utilization of the sparsity of neuromorphic computing, thereby enhancing energy efficiency. To the best of our knowledge, this is the first application of unstructured neuron pruning to deep SNNs. Experimental results demonstrate that our method achieves impressive energy efficiency gains. The sparse network pruned by our method with only 0.63% remaining connections can achieve a remarkable 91 times increase in energy efficiency compared to the original dense network, requiring only 8.5M SOPs for inference, with merely 2.19% accuracy loss on the CIFAR-10 dataset. Our work suggests that deep and dense SNNs exhibit high redundancy in energy consumption, highlighting the potential for targeted SNN sparsification to save energy. Codes are available at https://github.com/xyshi2000/Unstructured-Pruning.

## 1 Introduction

Spiking Neural Networks (SNNs), considered as the third generation of Artificial Neural Networks (ANNs) (Maass, 1997), have attracted significant attention on low-power edge computing with limited resources in recent years. In SNNs, the biologically-inspired neurons generate sparse and discrete events by firing binary spikes to communicate with postsynaptic neurons (Krestinskaya et al., 2019). Compared with ANNs, SNNs have the energy-saving advantage due to their event-driven nature when deployed on neuromorphic hardware (Furber et al., 2014; Merolla et al., 2014; Pei et al., 2019), and have become popular in neuromorphic computing in recent years (Schuman et al., 2017).

Recently, there has been a growing interest in developing deep SNN architectures capable of handling challenging tasks such as object recognition (Kim & Panda, 2021; Zhu et al., 2022), detection (Kim et al., 2020), text classfication (Lv et al., 2023), and robotic control (Tang et al., 2021). These architectures, such as SpikingVGG (Sengupta et al., 2019; Lee et al., 2020) and SpikingRes-Net (Zheng et al., 2021; Hu et al., 2021; Fang et al., 2021a), employ large-scale SNNs with numerous layers to achieve superior task performance (Duan et al., 2022). However, this comes at the cost of compromising the energy-saving benefits and necessitating greater computational resources.

Researchers make great efforts to enhance energy efficiency in SNNs. Since energy consumption of SNNs is closely tied to synaptic operations, existing methods aimed at improving energy efficiency primarily focus on two approaches: reducing the number of spikes or reducing the number of synaptic connections, i.e., reducing the model size. To reduce the number of spikes, several methods focus

---

*Corresponding author

on designing SNNs reasoning process to achieve spike number reduction at the algorithmic level, such as using sparse coding (Han & Roy, 2020; Rueckauer & Liu, 2018) or reducing the required reasoning time steps of the model (Ding et al., 2021; Bu et al., 2022; Chowdhury et al., 2022). Other algorithms specifically target spike reduction as an optimization goal (Kundu et al., 2021; Na et al., 2022). These efforts have shown promising results in reducing energy consumption. However, it should be noted that these methods do not fundamentally eliminate model redundancy.

Another technical route is to reduce the size of the model so as to eliminate redundancy. This approach involves various methods, including pruning (Chen et al., 2021; 2022), quantization (Wang et al., 2020; Chowdhury et al., 2021; Deng et al., 2021), and knowledge distillation (Kushawaha et al., 2021; Takuya et al., 2021; Tran et al., 2022; Xu et al., 2023). Quantization methods reduce the bitwise precision of the model parameters, while knowledge distillation methods transfer knowledge from a large model to a lightweight alternative. These methods reduce the model size in a structured manner, resulting in dense lightweight models. In comparison, fine-grained pruning shows a remarkable ability to eliminate redundancy in an unstructured manner and achieve high sparsity, thereby reducing synaptic connections without significant performance degradation. These researches suggest that pruning can significantly reduce energy consumption (Chowdhury et al., 2021).

While pruning is a promising energy reduction approach, previous methods have not directly optimized energy consumption. Existing pruning works have primarily been adapted from pruning methods designed for ANNs and focused more on enabling SNNs to run on hardware with limited computational resources. Consequently, these methods have a limited impact on energy consumption since they were not specifically designed for energy reduction purposes. Furthermore, most existing fine-grained pruning methods in ANNs primarily emphasize weight pruning, while the pruning of neurons is typically performed in a structured manner. When applied to SNNs, they overlook the fact that neurons in SNNs can also be pruned in a fine-grained manner. This oversight hinders the full utilization of the sparse computing capabilities of neuromorphic hardware.

This paper presents a comprehensive pruning framework for enhancing energy efficiency in SNNs. Our approach combines unstructured weight pruning with unstructured neuron pruning, leveraging the advantages of the sparse and event-driven nature of neuromorphic computing. By reducing energy consumption while maintaining low-performance loss, our method achieves a state-of-the-art balance between accuracy and energy efficiency. The main contributions are summarized as:

- We develop a model to analyze the energy consumption of SNNs and evaluate the effectiveness of weight and neuron pruning in reducing energy consumption.
- We propose a fine-grained pruning framework that integrates unstructured weight and neuron pruning to enhance the energy efficiency of SNNs. To the best of our knowledge, this is the first application of unstructured neuron pruning to deep SNNs.
- We present a novel design for the energy penalty term, addressing the ill-posed problem that arises when jointly pruning neurons and weights under energy constraints.
- Our method outperforms previous works in reducing energy consumption while maintaining comparable performance. The sparsest network pruned by our proposed method requires only 8.5M SOPs for inference, which is 91 times less than the original dense network, with merely 2.19% accuracy loss on the CIFAR10 dataset.

## 2 RELATED WORK

**Pruning methods for ANNs.** Pruning methods for ANNs can be broadly classified into three categories (Hoefler et al., 2021): 1) Unstructured weight pruning, 2) Structured weight pruning, and 3) Structured neuron pruning. Most unstructured weight pruning methods come from two starting points: heuristic strategies and training-aware methods. Heuristic methods, such as magnitude-based weight pruning (Han et al., 2015; Zhu & Gupta, 2018; Kusupati et al., 2020), are simple yet effective, and have garnered significant attention in early studies. Recent approaches mainly utilize training-aware methods, which leverage gradient-based criteria for pruning. These criteria are designed in diverse ways, such as modifying the behavior of forward propagation (Lin et al., 2020) or approximating $l_0$ regularization (Savarese et al., 2020). Different structured weight pruning methods exhibit varying levels of regularity, including stride-level, vector-level, kernel-level, and filter-level pruning (Mao et al., 2017). Previous studies (Wen et al., 2016; Anwar et al., 2017) have investigated the performance and efficiency of these methods across different regularity levels. In the case of

structured neuron pruning, the focus has primarily been on the channel level. Existing approaches include utilizing trainable channel selectors (Luo & Wu, 2020), evaluating importance scores of neurons (Yu et al., 2018), or applying regularization to the scaling factors of Batch Normalization (BN) layers (Zhuang et al., 2020). In the case of unstructured neuron pruning, several works dynamically "prune" neurons, i.e., "prune" out small activations (Kurtz et al., 2020; Sekikawa & Uto, 2021).

**Pruning methods for SNNs.** Recent research in SNN pruning mainly focuses on weight pruning, drawing inspiration from the well-established pruning methods developed for ANNs. Typical works include using the magnitude-based method (Yin et al., 2021), combining classic optimization tools with the SNN training method (Deng et al., 2021), etc. In addition, recent studies have explored bio-inspired pruning algorithms that leverage the similarities between SNNs and neural systems. These algorithms primarily focus on modeling the synaptic regrowth process, employing probability (Bellec et al., 2018) or gradient (Chen et al., 2021; 2022) as criteria for regrowth. In contrast to weight pruning, the study of neuron pruning for SNNs remains relatively limited. Wu et al. (2019) propose a method based on neuron similarity to select neurons for removal. However, this approach cannot be applied to SNNs with complex and deep structures, limiting its practicality.

**Other energy-saving methods for SNNs.** Apart from pruning, there are three main energy-saving methods: 1) Quantization, 2) Reduction of firing rates, and 3) Knowledge distillation. Quantization methods aim to enhance efficiency by reducing the precision of weights (Deng et al., 2021) to match the reduced bit precision of neuromorphic hardware platforms (Qiao et al., 2021; Stromatias et al., 2015). Recent research on quantization methods for SNNs mainly focuses on weight binarization (Qiao et al., 2021; Kheradpisheh et al., 2022; Wang et al., 2020). Directly reducing the firing rate of neurons helps decrease computational overhead. Previous approaches introduce a penalty term for firing rate as a form of regularization (Deng et al., 2021; Neil et al., 2016), or employ network architecture search methods (Na et al., 2022) to discover SNN structures with low firing rates. Knowledge distillation explores transferring knowledge from large-scale ANNs (Takuya et al., 2021; Tran et al., 2022; Xu et al., 2023; Guo et al., 2023) or SNNs (Kushawaha et al., 2021) to smaller-scale SNNs to compress the models and reduce energy consumption.

## 3 MOTIVATION

In this section, we first model the energy consumption of SNNs. Based on this model, we analyze the limits of existing pruning methods of SNNs and point out that pruning neurons and weights in SNNs contribute to balancing energy efficiency and performance.

### 3.1 ENERGY CONSUMPTION MODEL

The primary metric used to assess the energy consumption of neuromorphic chips is the average energy required to transmit a single spike through a synapse (Furber, 2016). This is due to the significant energy cost associated with synapse processing, making it a crucial factor in overall energy consumption. For hardware-independent theoretical analysis, we treat the energy consumption of the entire system corresponding to one synaptic operation[1] (SOP) on average as a known constant, and use the SOPs as the metric of energy consumption of the model. Specifically, we employ the following model to estimate the energy consumption of an SNN:

$$E = C_E \cdot \#SOP = C_E \sum_i s_i c_i, \tag{1}$$

where $C_E$ denotes the energy consumption of one SOP, $\#SOP = \sum_i s_i c_i$ denotes the total number of synaptic operations. For each presynaptic neuron $i$ in an SNN, $s_i$ denotes the number of spikes fired by this neuron, and $c_i$ denotes the number of synaptic connections from this presynaptic neuron. It's worth noting that such a linear energy model may not be suitable for all hardware architectures. We believe that it matches the architectures that are highly optimized for sparsity. A detailed discussion of the suitability can be found in the appendix.

**Energy consumption model for sparse SNNs.** For a sparse SNN, we restate Eq. (1) as follows:

$$E = C_E \sum_i \left( s_i \sum_j n_i^{pre} \wedge \theta_{ij} \wedge n_{ij}^{post} \right). \tag{2}$$

---

[1] In this paper, we define SOP as the operations performed when a spike passes through a synapse.

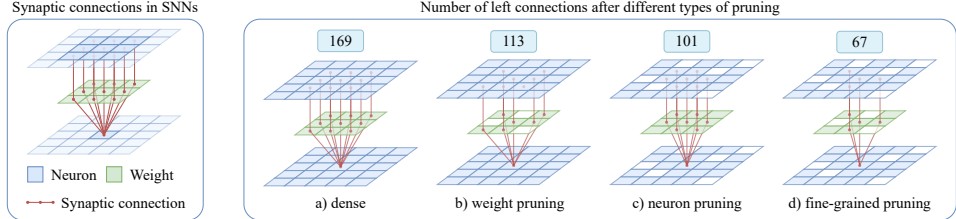

Figure 1: Diagram of synaptic connections in SNNs and number of connections left after pruning. **Left**: Neurons, weights, and synaptic connections in a convolutional layer with $5 \times 5$ input and a $3 \times 3$ filter. **Right**: The effect of different kinds of pruning on the number of synaptic connections. The number on the top of each subfigure indicates the number of synaptic connections left.

Here, for each synaptic connection from $i$-th presynaptic neuron to its $j$-th postsynaptic neuron, $(n_i^{pre}, \theta_{ij}, n_{ij}^{post}) \in \{0,1\} \times \{0,1\} \times \{0,1\}$ denotes the states of presynaptic neuron, synapse, and postsynaptic neuron. 1 means alive, while 0 means pruned. $\wedge$ is the logical AND operation.

## 3.2 Benefits of Fine-grained SNN Pruning

Eq. (2) illustrates that presynaptic neurons, synapses, and postsynaptic neurons all contribute to the computation in SNNs. When any of these components are pruned, the corresponding connections are also pruned, eliminating the need for computations associated with those connections. Fig. 1 provides an example of a simple convolution operation with a 3×3 filter and a 5×5 input with zero-padding, demonstrating how fine-grained pruning significantly reduces synaptic connections.

While both neuron pruning and weight pruning are valid approaches, existing SNN pruning methods focus more on weight pruning, and neuron pruning is only performed in a structured manner or on shallow fully-connected networks (Wu et al., 2019). The main concern is that in shallow fully-connected networks, neuron pruning can effectively be replaced by weight pruning since pruning a neuron is equivalent to eliminating all associated weights. However, this approach is not applicable to convolutional layers in SNNs. In convolutional operations, the convolution kernel weights are multiplied with each element in the input channel (ignoring stride). Pruning all weights associated with a neuron would result in the entire channel being pruned. Therefore, weight pruning fails to capture the unstructured nature of neuron pruning, or in other words, is not fine-grained enough.

Except for structural considerations, fine-grained unstructured neuron pruning offers a better trade-off between performance and energy efficiency. The spatial correlation among features in convolutional layers means that neighboring elements often carry similar information. Consequently, unstructured neuron pruning, when applied at an appropriate sparsity level, does not compromise performance. This enables further reduction in energy consumption while maintaining comparable performance.

## 4 Method

In this section, we first introduce the spiking neuron model in SNNs, and then present a general formulation of the unstructured pruning framework with energy consumption constraint. Finally, we delve into the design of the energy constraint and derive the comprehensive minimization problem.

### 4.1 Spiking Neuron Model

Similar to Fang et al. (2021b), we use the following unified discrete-time model to describe the dynamics of spiking neurons:

$$v[t] = g(u[t], \sum_i w_i I_i[t]), \tag{3a}$$

$$o[t] = H(v[t] - u_{\text{th}}), \tag{3b}$$

$$u[t+1] = o[t]u_{\text{rest}} + (1 - o[t])v[t]. \tag{3c}$$

Here $I_i[t]$ denotes the input current from the $i$-th presynaptic neuron at time-step $t$, and $w_i$ is the corresponding synaptic weight. $v[t]$ and $u[t+1]$ denote the membrane potential of the neuron before and after firing at time-step $t$. $H(\cdot)$ is the Heaviside step function. $o[t] \in \{0, 1\}$ denotes whether or not the neuron fires a spike at time-step $t$. When the membrane potential $v[t]$ at time step $t$ reaches the firing threshold $u_{\text{th}}$, the neuron will fire a spike, and the membrane potential after firing $u[t+1]$ will reset to the resting potential $u_{\text{rest}}$. The function $g(\cdot, \cdot)$ describes the neuronal dynamics and differs between different spiking neuron models. For example, the Leaky Integrate-and-Fire (LIF) model can be defined by Eq. (4):

$$g(u[t], x[t]) = u[t] + \frac{1}{\tau}(x[t] - (u[t] - u_{\text{rest}})), \tag{4}$$

where $\tau$ denotes the membrane time constant and $x[t]$ denotes the input.

## 4.2 Pruning of Weights and Neurons

In order to describe the sparse spiking subnetwork, we introduce binary masks $m_n$ and $m_w$ to indicate the state of the neurons and weights, respectively. Given a neuron, we restate Eq. (3a) with binary masks as follows:

$$v[t] = g(u[t], \sum_i (m_{w,i}\, w_i)(m_{n,i}\, I_i[t])), \tag{5}$$

where $(m_{n,i}, m_{w,i}) \in \{0, 1\} \times \{0, 1\}$ are the binary masks of the $i$-th presynaptic neuron and the corresponding synaptic weight. A mask value of 0 indicates that the presynaptic neuron or synaptic weight is pruned, while a value of 1 indicates that it is active.

## 4.3 Sparse Structure Learning with Energy Constraints

Given a spiking neural network $f(\cdot)$ with weights $\boldsymbol{w} \in \mathbb{R}^{d_w}$ and masks of weights and neurons $\boldsymbol{m}_w \in \{0, 1\}^{d_w}$, $\boldsymbol{m}_n \in \{0, 1\}^{d_n}$, where $d_n$ and $d_w$ denote the number of neurons and weights of network $f$, respectively, we formulate the learning of the sparse SNN as a loss minimization problem with energy constraints as follows:

$$\arg \min_{\boldsymbol{w}, \boldsymbol{m}_w, \boldsymbol{m}_n} \mathcal{L}(f(\cdot; \boldsymbol{m}_w \odot \boldsymbol{w}, \boldsymbol{m}_n)) + \lambda E(\cdot; f, \boldsymbol{m}_w, \boldsymbol{m}_n). \tag{6}$$

Here $\boldsymbol{m}_w \in \{0, 1\}^{d_w}$ and $\boldsymbol{m}_n \in \{0, 1\}^{d_n}$ denote the masks of weights and neurons, respectively. $\odot$ denotes element-wise multiplication. $\mathcal{L}(f(\cdot; \boldsymbol{m}_w \odot \boldsymbol{w}, \boldsymbol{m}_n))$ denotes the loss of the prediction of model $f(\cdot; \boldsymbol{m}_w \odot \boldsymbol{w}, \boldsymbol{m}_n)$. $E(\cdot; f, \boldsymbol{m}_w, \boldsymbol{m}_n)$ is the energy consumption penalty term, corresponding to the estimated energy consumption of the network $f(\cdot; \boldsymbol{m}_w \odot \boldsymbol{w}, \boldsymbol{m}_n)$ based on the model presented in Eq. (2). The parameter $\lambda$ controls the trade-off between performance and energy consumption.

To address the non-differentiability of the binary mask $m$, we re-parameterize $m$ by introducing a new parameter $\alpha \in \mathbb{R}$ to approximate $m$. Specifically, we define $m = H(\alpha)$, where $H(\cdot)$ represents the Heaviside step function. As $H(\cdot)$ is also non-differentiable, we propose using the scaled sigmoid function $\sigma(\alpha; \beta) = \sigma(\beta\alpha) = \frac{1}{1+e^{-\beta\alpha}}$ to approximate the Heaviside step function $H(\cdot)$. The scaled sigmoid function is smooth and approaches the Heaviside step function as $\beta$ tends to infinity, that is, $\lim_{\beta \to \infty} \sigma(\alpha; \beta) = H(\alpha) = m$. By re-parameterizing the problem in this way, we can reformulate the original minimization problem of Eq. (6) as follows:

$$\arg \min_{\boldsymbol{w}, \boldsymbol{\alpha}_w, \boldsymbol{\alpha}_n} \lim_{\beta \to \infty} \mathcal{L}(f(\cdot; \sigma(\beta\boldsymbol{\alpha}_w) \odot \boldsymbol{w}, \sigma(\beta\boldsymbol{\alpha}_n))) + \lambda E(\cdot; f, \sigma(\beta\boldsymbol{\alpha}_w), \sigma(\beta\boldsymbol{\alpha}_n)). \tag{7}$$

To achieve an effective approximation of the binary mask $m$, scale factor $\beta$ must be large enough. However, using a fixed large value for $\beta$ can result in model degradation. In our experiments, we adopt a gradual scheduling approach for the scale factor $\beta$, transitioning it from an initial value $\beta_0$ to a final value $\beta_T$ using the exponential function $\beta(t) = \beta_0 \left(\frac{\beta_T}{\beta_0}\right)^{\frac{t}{T_s}}$, where $T_s$ represents the number of training epochs. After training the weights and masks for $T_s$ epochs, we replace the scaled sigmoid function with the Heaviside step function and continue training with fixed masks. An overview of the proposed method is depicted in Fig. 2.

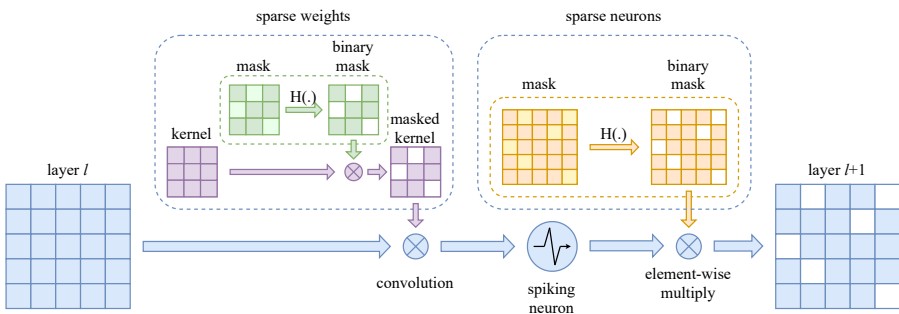

Figure 2: Overview of our framework.

## 4.4 DESIGN OF ENERGY CONSUMPTION PENALTY TERM

Eq. (7) represents the general form of the minimization problem, incorporating an energy consumption penalty term. However, it does not provide specific details regarding the implementation of the penalty term. In this section, we will delve into the design of the penalty term. According to Eqs. (2) and (7), the penalty term can be defined as follows:

$$E = C_E \sum_i s_i \sum_j \sigma(\beta\alpha_{n,i}^{\text{pre}}) \cdot \sigma(\beta\alpha_{w,ij}) \cdot \sigma(\beta\alpha_{n,ij}^{\text{post}}), \tag{8}$$

where $\alpha_{n,i}^{\text{pre}}, \alpha_{w,ij}, \alpha_{n,ij}^{\text{post}}$ are the re-parameterized mask values of $n_i^{\text{pre}}, \theta_{ij}, n_{ij}^{\text{post}}$, respectively. Note that here we replace the logical AND operation with multiplication to facilitate easier optimization.

This penalty term is directly derived from the energy consumption model of sparse SNNs by substituting the continuous masks into Eq. (2). While it provides an intuitive formulation, optimizing this term can be challenging. The main difficulty arises from the multiplication of the three continuous masks, which makes the problem ill-posed. Consequently, the solution becomes highly sensitive to the initial values of hyperparameters such as the scale factor $\beta_0$ and the reparameterized mask $\alpha_0$. Improper initialization can easily lead to degradation, i.e., one type of elements (neurons or weights) is completely pruned, resulting in a trivial solution to the minimization problem.

To solve this problem, we convert this composite constraint term into two independent sub-constraints by rewriting Eq. (8) as follows:

$$E_n = C_E \sum_i \sigma(\beta\alpha_{n,i}^{\text{pre}}) \left( s_i \sum_j \sigma(\beta\alpha_{w,ij}) \cdot \sigma(\beta\alpha_{n,ij}^{\text{post}}) \right) = C_E \sum_i e_{n,i}\sigma(\beta\alpha_{n,i}^{\text{pre}}), \tag{9a}$$

$$E_w = C_E \sum_p \sigma(\beta\alpha_{w,p}) \left( \sum_q s_{pq} \cdot \sigma(\beta\alpha_{n,pq}^{\text{pre}}) \cdot \sigma(\beta\alpha_{n,pq}^{\text{post}}) \right) = C_E \sum_p e_{w,p}\sigma(\beta\alpha_{w,p}). \tag{9b}$$

In Eq. (9a), we first place the presynaptic neuron mask $\sigma(\beta\alpha_n^{\text{pre}})$ in the outer summation as each spike fired by the presynaptic neuron associated with itself. For Eq. (9b), since the total number of SOPs is independent of the counting method, we adopt an alternative counting approach. Specifically, we count the number of spikes transmitted through the synapses associated with each weight. In this context, for the $p$-th weight, $\sigma(\beta\alpha_{w,p})$ represents its mask, while $s_{pq}$, $\sigma(\beta\alpha_{n,pq}^{\text{pre}})$, and $\sigma(\beta\alpha_{n,pq}^{\text{post}})$ denote the number of transmitted spikes, the mask of the presynaptic neuron, and the mask of the postsynaptic neuron, respectively, for the $q$-th synapse associated with this weight.

To obtain independent sub-constraints, we simplify the dynamic nature of connections by approximating the number of connections associated with each element as a constant. Specifically, in Eq. (9a), we focus on the presynaptic neuron and treat $\sigma(\beta\alpha_{w,ij})$ and $\sigma(\beta\alpha_{n,ij}^{\text{post}})$ as constant. Similarly, in Eq. (9b), we focus on synaptic weight and treat $\sigma(\beta\alpha_{n,pq}^{\text{pre}})$ and $\sigma(\beta\alpha_{n,pq}^{\text{post}})$ as constant. Note that in Eq. (9a), we specifically focus on the SOPs associated with presynaptic neurons, since a neuron in SNNs typically functions as both a presynaptic and postsynaptic neuron, and accurately counting the SOPs associated with postsynaptic neurons proves to be challenging. To further

Table 1: Performance under different sparsity levels

| Dataset | Arch. | $\lambda$ | Top-1 Acc. (%) | Acc. Loss (%) | Avg. SOPs (M) | Ratio | Conn. (%) | Neuron (%) | Weight (%) |
|---|---|---|---|---|---|---|---|---|---|
| CIFAR10 | 6 Conv, 2 FC | (dense) | 92.84 | 0 | 776.12 | 1 | 100 | 100 | 100 |
| | | $5 \times 10^{-12}$ | 92.63 | -0.21 | 38.32 | 20.25 | 2.97 | 21.03 | 34.24 |
| | | $2 \times 10^{-11}$ | 92.05 | -0.79 | 16.47 | 47.12 | 1.16 | 12.27 | 26.05 |
| | | $3 \times 10^{-11}$ | 91.47 | -1.37 | 11.98 | 64.78 | 0.87 | 10.31 | 23.00 |
| | | $5 \times 10^{-11}$ | 90.65 | -2.19 | 8.50 | 91.31 | 0.63 | 8.23 | 19.34 |
| DVS-CIFAR10 | VGGSNN | (dense) | 82.4 | 0 | 647.30 | 1 | 100 | 100 | 100 |
| | | $1 \times 10^{-10}$ | 81.9 | -0.5 | 47.81 | 13.54 | 6.80 | 42.73 | 36.35 |
| | | $2 \times 10^{-10}$ | 81.0 | -1.4 | 31.86 | 20.32 | 4.46 | 35.13 | 33.02 |
| | | $7 \times 10^{-10}$ | 79.0 | -3.4 | 10.02 | 64.60 | 1.27 | 18.56 | 22.56 |
| | | $1 \times 10^{-9}$ | 78.3 | -4.1 | 6.75 | 95.90 | 0.77 | 14.11 | 18.66 |
| ImageNet | SEW ResNet18 | (dense) | 63.18 | 0 | 1337.87 | 1 | 100 | 100 | 100 |
| | | $5 \times 10^{-11}$ | 61.89 | -1.29 | 311.70 | 4.29 | 28.60 | 74.86 | 35.26 |
| | | $1 \times 10^{-10}$ | 60.00 | -3.18 | 177.99 | 7.52 | 19.31 | 68.27 | 27.74 |
| | | $2 \times 10^{-10}$ | 58.99 | -4.19 | 116.88 | 11.45 | 14.19 | 51.17 | 23.52 |

simplify this term, we assume that the average firing rate remains constant. Thus, the number of transmitted spikes on each synapse $s_i$ and $s_{pq}$ can be regarded as constants. Based on these approximations, we treat $e_{n,i} = s_i \sum_j \sigma(\beta\alpha_{w,ij}) \cdot \sigma(\beta\alpha_{n,ij}^{\text{post}})$ and $e_{w,p} = \sum_q s_{pq} \cdot \sigma(\beta\alpha_{n,pq}^{\text{pre}}) \cdot \sigma(\beta\alpha_{n,pq}^{\text{post}})$ as constant, here $e_i$ and $e_p$ denote the approximate SOPs associated with $i$-th neuron and $p$-th weight, respectively. Furthermore, given that the sigmoid function and SOPs are always positive, we observe that this penalty term is equivalent to the $l_1$ regularization.

$$E_n = C_E \|e_n \sigma(\beta\boldsymbol{\alpha}_n)\|_1, \quad E_w = C_E \|e_w \sigma(\beta\boldsymbol{\alpha}_w)\|_1, \tag{10}$$

where we use the vector form of $\boldsymbol{e}_n$ and $\boldsymbol{e}_w$ to replace the scalar form of $e_{n,i}$ and $e_{w,p}$. By applying Eq. (10) to Eq. (7), the final form of the minimization problem is as follows:

$$\arg\min_{\boldsymbol{w},\boldsymbol{\alpha}_w,\boldsymbol{\alpha}_n} \lim_{\beta\to\infty} \mathcal{L}(f(\cdot; \sigma(\beta\boldsymbol{\alpha}_w) \odot \boldsymbol{w}, \sigma(\beta\boldsymbol{\alpha}_n))) + \lambda\|\boldsymbol{e}_w\sigma(\beta\boldsymbol{\alpha}_w)\|_1 + \lambda\|\boldsymbol{e}_n\sigma(\beta\boldsymbol{\alpha}_n)\|_1, \tag{11}$$

Note that the energy constant $C_E$ is omitted here, as it can be absorbed into the parameter $\lambda$ during optimization. The entire learning process is summarized in Algorithm 1 of the appendix.

## 5 EXPERIMENTS

In this section, we evaluate the effectiveness of our method for classification tasks on the CIFAR-10 (Krizhevsky et al., 2009), DVS-CIFAR10 (Li et al., 2017), and ImageNet (Deng et al., 2009) datasets. We first analyze the performance under different sparsity levels, including accuracy and energy consumption. Further, we compare our method with other state-of-the-art approaches. Finally, we examine the effectiveness of unstructured neuron pruning and the combination with weight pruning by ablation study. More details of experimental settings can be found in the appendix.

### 5.1 PERFORMANCE UNDER DIFFERENT SPARSITY LEVELS

We analyze the test accuracy and energy consumption under different sparsity levels and report the results in Tab. 1. Here "Avg. SOPs" denotes the average SOPs required for an instance in inference. This value is used to estimate energy consumption, as discussed in Sec. 3.1. Additionally, "Ratio" represents the energy saving ratio compared to the original dense network. "Conn.", "Neuron", and "Weight" refer to the percentage of remaining connections, neurons, and weights, respectively.

For the CIFAR-10 dataset, we adjust the parameter $\lambda$ from $5 \times 10^{-12}$ to $5 \times 10^{-11}$ to control the sparsity level. The sparsest network achieved, with $\lambda = 5 \times 10^{-11}$, retains only 0.63% of the original connections. Remarkably, a single instance now requires only 8.5M SOPs for inference, which is 91.31 times less than the original dense network, with merely 2.19% decrease in test accuracy. Even for smaller values of $\lambda$, our method remains effective in reducing synaptic connections and energy consumption. For instance, we achieve 20.25 times more energy efficiency than the original dense network with only 2.97% connections and 0.21% accuracy loss. These results demonstrate the effectiveness of our method in discovering sparse networks with remarkably low energy consumption. Additional experimental results on the CIFAR-100 dataset can be found in Tab. A1 in the appendix.

Table 2: Performance comparison between the proposed method and the state-of-the-art methods

| Dataset | Pruning Method | Arch. | T | Top-1 Acc. (%) | Acc. Loss (%) | Avg. SOPs (M) | Ratio | Conn.[2] (%) | Param. (M) |
|---|---|---|---|---|---|---|---|---|---|
| CIFAR10 | ADMM[1] | 7 Conv, 2 FC | 8 | 90.19 | -0.13 | 107.97 | 2.91 | 25.03 | 15.54 |
| | | | | 88.18 | -2.14 | 49.72 | 6.32 | 10.34 | 6.22 |
| | | | | 79.63 | -10.69 | 28.65 | 10.96 | 5.35 | 3.11 |
| | Grad R | 6 Conv, 2 FC | 8 | 92.54 | -0.30 | 371.05 | 2.09 | 36.72 | 10.43 |
| | | | | 92.50 | -0.34 | 232.51 | 3.33 | 22.24 | 4.42 |
| | | | | 92.03 | -0.81 | 143.69 | 5.40 | 11.85 | 1.86 |
| | | | | 91.37 | -1.47 | 87.73 | 8.85 | 6.24 | 0.86 |
| | | | | 89.32 | -3.52 | 41.89 | 18.53 | 2.15 | 0.26 |
| | ESLSNN[1] | ResNet19 | 2 | 92.08 | -0.63 | 298.24 | 2.11 | 32.71 | 2.53 |
| | | | | 91.46 | -1.25 | 178.10 | 3.54 | 16.06 | 1.26 |
| | | | | 90.90 | -1.81 | 108.89 | 5.79 | 8.21 | 0.63 |
| | STDS | 6 Conv, 2 FC | 8 | 92.49 | -0.35 | 147.22 | 5.27 | 11.33 | 1.71 |
| | | | | 92.49 | -0.35 | 87.94 | 8.83 | 5.29 | 0.82 |
| | | | | 91.64 | -1.20 | 56.49 | 13.74 | 2.98 | 0.51 |
| | | | | 91.06 | -1.78 | 37.16 | 20.89 | 1.91 | 0.36 |
| | | | | 90.21 | -2.63 | 26.81 | 28.95 | 1.35 | 0.28 |
| | **This work** | 6 Conv, 2 FC | 8 | **92.63** | **-0.21** | **38.32** | **20.25** | **2.97** | **12.57** |
| | | | | **92.05** | **-0.79** | **16.47** | **47.12** | **1.16** | **9.56** |
| | | | | **91.47** | **-1.37** | **11.98** | **64.78** | **0.87** | **8.44** |
| | | | | **90.65** | **-2.19** | **8.50** | **91.31** | **0.63** | **7.10** |
| DVS-CIFAR10 | ESLSNN[1] | VGGSNN | 10 | 80.6 | -1.8 | 266.17 | 2.43 | 36.19 | 1.94 |
| | | | | 79.2 | -3.2 | 152.64 | 4.24 | 17.6 | 0.97 |
| | | | | 77.5 | -4.9 | 79.65 | 8.13 | 8.51 | 0.48 |
| | STDS[1] | VGGSNN | 10 | 81.7 | -0.7 | 225.34 | 2.87 | 21.36 | 2.07 |
| | | | | 81.1 | -1.3 | 76.56 | 8.45 | 10.17 | 0.65 |
| | | | | 79.8 | -2.6 | 38.85 | 16.66 | 4.67 | 0.24 |
| | **This work** | VGGSNN | 10 | **81.9** | **-0.5** | **47.81** | **13.54** | **6.80** | **3.52** |
| | | | | **81.0** | **-1.4** | **31.86** | **20.32** | **4.46** | **2.50** |
| | | | | **79.0** | **-3.4** | **10.02** | **64.60** | **1.27** | **2.18** |
| | | | | **78.3** | **-4.1** | **6.75** | **95.90** | **0.77** | **1.81** |
| ImageNet | STDS | SEW ResNet18 | 4 | 62.84 | -0.34 | 1082.60 | 1.24 | 77.54 | 7.42 |
| | | | | 61.51 | -1.67 | 512.95 | 2.61 | 34.79 | 2.38 |
| | | | | 61.30 | -1.88 | 424.07 | 3.15 | 29.29 | 1.94 |
| | | | | 59.93 | -3.25 | 273.39 | 4.89 | 21.08 | 1.24 |
| | | | | 58.06 | -5.12 | 182.49 | 7.33 | 15.71 | 0.75 |
| | | | | 56.28 | -6.90 | 134.64 | 9.94 | 12.95 | 0.52 |
| | **This work** | SEW ResNet18 | 4 | **61.89** | **-1.29** | **311.70** | **4.29** | **28.60** | **3.93** |
| | | | | **60.00** | **-3.18** | **177.99** | **7.52** | **19.31** | **3.10** |
| | | | | **58.99** | **-4.19** | **116.88** | **11.45** | **14.19** | **2.62** |

[1] Our implementation
[2] The definition of "connections" in this paper is different from GradR and STDS. Here Conn. refers to percentage of synaptic connections left, whereas "connections" in GradR and STDS refer to the weights left, corresponding to Param.

For the DVS-CIFAR10 dataset, our method achieves a top-1 accuracy of 81.9% (merely 0.5% accuracy loss) and is 13.54 times more energy efficient than the original dense network when constrained to 6.80% connectivity. Moreover, we achieve 95.90 times more energy efficiency than the original dense network with only 0.77% connections and 4.1% accuracy loss. These results demonstrate that our method also applies to neuromorphic datasets.

Additionally, we evaluated the performance of our method on a large-scale dataset, namely, the ImageNet dataset. The ratio of energy efficiency gains is lower compared to the CIFAR10 and DVS-CIFAR10 datasets. This is because ImageNet classification is more challenging, and SEW ResNet18 is not a large model, resulting in less redundancy available for pruning. Nevertheless, our method still achieves 4.29 times more energy efficiency than the original SEW ResNet18 with 1.29% top-1 accuracy loss, and 11.45 times more efficiency with 4.19% accuracy loss.

## 5.2 COMPARISON WITH THE STATE-OF-THE-ART

To evaluate the effectiveness of our method, we compare it with other state-of-the-art SNN algorithms ADMM (Deng et al., 2021), Grad R (Chen et al., 2021), ESLSNN (Shen et al., 2023) and STDS (Chen et al., 2022). We present the results in Tab. 2. Since these works did not provide the average SOPs required for inference on the test set, we evaluate these values using our implementation. Additionally, we visualized the comparative results for the CIFAR-10 dataset in Fig. 3.

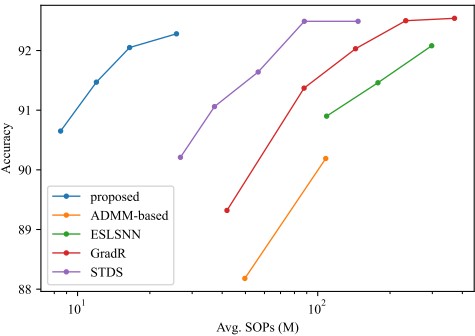 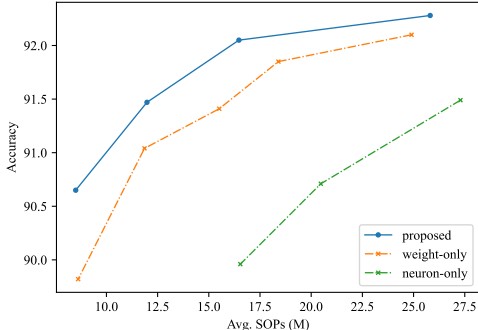

Figure 3: Comparison with the state-of-the-art      Figure 4: Results of ablation study

As shown in Tab. 2 and Fig. 3, our method achieves comparable performance while significantly reducing energy consumption compared to the state-of-the-art methods. For example, on the CIFAR-10 dataset, the current leading method, STDS, achieves a top-1 accuracy of 90.21% with 1.35% connections and 26.81M SOPs. In contrast, our method achieves a higher accuracy of 90.65% with only 0.63% connections and 8.5M SOPs. In Fig. 3, our data line is positioned in the upper left corner of all other methods, indicating that our proposed method achieves lower SOPs with equal accuracy.

Furthermore, we observe an interesting phenomenon. Fewer weights left does not necessarily lead to lower energy consumption. As indicated in Tab. 2, GradR achieves fewer weights than STDS (0.26M vs. 0.28M) using the same dense network structure. However, GradR requires an average of 41.89M SOPs, whereas STDS only requires 26.81M SOPs. This discrepancy arises from the fact that the model pruned by GradR has higher connectivity (2.15% vs. 1.35%). According to our energy consumption model in Sec. 3.1, assuming the average firing rate of each neuron remains the same, energy consumption is directly proportional to the connectivity. Therefore, this phenomenon validates our proposed energy consumption model that the SOPs is proportional to the connectivity.

### 5.3 ABLATION STUDY

To validate the effectiveness of combining unstructured neuron pruning with weight pruning, we conduct experiments where we individually prune neurons or weights and compare their performance with our proposed method on the CIFAR-10 dataset. For a fair comparison, all experimental settings remain the same, except for the prunable elements and the weight factor of the penalty term $\lambda$. As shown in Fig. 4, our proposed method outperforms pruning only one type of element. Notably, the performance difference becomes more significant as the sparsity level increases. Detailed experimental results are listed in Tab. A2 in the appendix.

Upon further analysis of the results, we identify the limitations of pruning only one type of element. Pruning weights could achieve comparable or slightly worse performance at low sparsity levels, while still resulting in high energy consumption. It proves ineffective in further reducing energy consumption. For instance, as sparsity increases, the accuracy drops by 1.22%, while SOPs decrease by a mere 3.24M. On the other hand, pruning neurons compromises performance but offers a substantial reduction in energy consumption with less accuracy loss. Therefore, combining neuron and weight pruning represents a trade-off between accuracy and energy consumption, enabling a more effective reduction in energy consumption while maintaining acceptable performance levels.

## 6 CONCLUSION

In this paper, we propose a novel framework that combines unstructured weight and neuron pruning to enhance the energy efficiency of SNNs. We detail the design of the energy consumption constraint to overcome the ill-posed problem. Experimental results demonstrate the efficacy of our approach in significantly reducing the energy consumption of deep SNNs, achieving higher energy efficiency than previous methods while maintaining comparable performance. Our work highlight the immense potential of SNN sparsification methods that target energy-saving.

## ACKNOWLEDGMENTS

This work was supported by the National Natural Science Foundation of China (62176003, 62088102) and by Beijing Nova Program (20230484362).

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

# A  APPENDIX

## A.1  LEARNING PROCESS

The learning process of our proposed method, utilizing stochastic gradient descent (SGD), is summarized in Algorithm 1. It should be noted that this algorithm can be adapted to other optimizing strategies, such as SGD with momentum or Adam, with straightforward modifications.

To enhance readability, we utilize simplified notation in Algorithm 1, representing specific terms as $\mathcal{L}_c$, $\mathcal{L}_s$, and $\mathcal{E}$.

$$\mathcal{L}_c(x) = \mathcal{L}(f(x; \sigma(\beta\boldsymbol{\alpha}_w) \odot \boldsymbol{w}, \sigma(\beta\boldsymbol{\alpha}_n))), \tag{A1a}$$

$$\mathcal{L}_s(x) = \mathcal{L}(f(x; H(\boldsymbol{\alpha}_w) \odot \boldsymbol{w}, H(\boldsymbol{\alpha}_n))), \tag{A1b}$$

$$\mathcal{E} = \|\boldsymbol{e}_w \sigma(\beta\boldsymbol{\alpha}_w)\|_1 + \|\boldsymbol{e}_n \sigma(\beta\boldsymbol{\alpha}_n)\|_1. \tag{A1c}$$

Here, $\mathcal{L}_c(\cdot)$ denotes the loss of network $f$ with continuous masks, $\mathcal{L}_s(\cdot)$ denotes the loss of network $f$ with binary masks, $x$ is the mini-batch input, and $\mathcal{E}$ corresponds to the penalty term.

We also review the meaning of some symbols: $\boldsymbol{e}_w$ and $\boldsymbol{e}_n$ are constants, $\boldsymbol{w}$ is the parameters of network $f$, $\boldsymbol{\alpha}_w$ and $\boldsymbol{\alpha}_n$ are the re-parameterized masks of weights and neurons, respectively, $\sigma(\cdot)$ is the sigmoid function, $\beta$ is the scale factor of the sigmoid function, $H(\cdot)$ is the Heaviside step function, $\odot$ is the element-wise multiplication operation.

## A.2  INTRODUCTION AND PREPROCESSING OF DATASETS

**CIFAR-10**    The CIFAR-10 dataset (Krizhevsky et al., 2009) is a static image dataset. It contains 60,000 samples divided into 10 categories, with 6,000 samples per category. Each category consists of 5,000 training samples and 1,000 testing samples. Each image in the dataset has a resolution of $32 \times 32$ pixels and is in color.

We perform data normalization to ensure that the inputs have zero mean and unit variance. Additionally, we use the common data augmentation similar to those used in previous works (Chen et al., 2021; 2022; Deng et al., 2021; Shen et al., 2023; Zhu et al., 2023), i.e., random horizontal flipping and random cropping with a padding of 4, to avoid over-fitting.

---

**Algorithm 1:** Learning Process

---

**Input:** Network $f$, parameters $w$, re-parameterized masks $\alpha_w, \alpha_n$, weight of penalty term $\lambda$, learning rate $\eta$, initial scale factor $\beta_0$, final scale factor $\beta_T$, pruning epochs $T_s$, fine-tuning epochs $T_f$, number of batches of training set $N$

**Output:** Sparse network $f(\cdot; H(\alpha_w) \odot w, H(\alpha_n))$

---

```
// Initialization
```
Calculate $e_w, e_n$ of each weight and neuron.
Initialize $w, \alpha_w, \alpha_n$;
Initialize $\beta = \beta_0$;
```
// Pruning Phase
```
**for** $i = 0$ to $T_s$ **do**
    **foreach** batch $x$ in training set **do**
        Update parameters $w \leftarrow w - \eta \nabla_w \mathcal{L}_c(x)$;
        Update weight masks $\alpha_w \leftarrow \alpha_w - \eta(\nabla_{\alpha_w} \mathcal{L}_c(x) + \lambda \cdot \nabla_{\alpha_w} \mathcal{E})$;
        Update neuron masks $\alpha_n \leftarrow \alpha_n - \eta(\nabla_{\alpha_n} \mathcal{L}_c(x) + \lambda \cdot \nabla_{\alpha_n} \mathcal{E})$;
        Update scale factor $\beta \leftarrow \beta \cdot (\frac{\beta_T}{\beta_0})^{1/(N \cdot T_s)}$;

```
// Fine-tuning Phase
```
**for** $i = 0$ to $T_f$ **do**
    **foreach** batch $x$ in training set **do**
        Update parameters $w \leftarrow w - \eta \nabla_w \mathcal{L}_s(x)$;

---

**DVS-CIFAR10** The DVS-CIFAR10 dataset (Li et al., 2017) is a neuromorphic dataset consisting of event streams, providing rich temporal information. It consists of 10,000 samples divided into 10 categories, with 1,000 samples per category. Each sample is an event stream with a size of $128 \times 128$. Since the DVS-CIFAR10 does not separate samples into the training set and test set, we choose the first 900 samples of each category for training and the last 100 for testing.

For each sample in the DVS-CIFAR10 dataset, we split the event stream into 10 equal slices, each containing the same number of events. Then, for each slice, we integrate the events into a single frame and use this frame as the input on that particular time step. Consequently, the time step we use on DVS-CIFAR10 is 10.

Compared to static datasets, neuromorphic datasets typically contain more noise. As a result, these datasets are more susceptible to over-fitting than static datasets, particularly for well-trained SNNs. To mitigate overfitting, we adopt the data augmentation technique proposed in (Li et al., 2022), which is also employed by ESLSNN (Shen et al., 2023). For a fair comparison, we apply this data augmentation approach to STDS (Chen et al., 2022) in our implementation as well.

**ImageNet** The ImageNet dataset (Deng et al., 2009) is a large-scale dataset widely used in computer vision tasks. It consists of approximately 1.2 million high-resolution images, categorized into 1,000 distinct classes, each containing roughly 1,000 images. Each class contains approximately 1,000 images, and encompasses a diverse range of objects and scenes, thus reflecting real-world scenarios effectively.

We use the standard preprocessing, i.e., data normalization, randomly crop and resize the images to $224 \times 224$, and random horizontal flipping for data augmentation.

**CIFAR-100** The CIFAR-100 dataset is an extension of the CIFAR-10 dataset, designed to provide a more challenging and diverse benchmark for image recognition algorithms. It contains 100 different classes or categories for classification, encompassing a wide range of objects and concepts compared to the CIFAR-10 dataset's limited set of 10 classes.

For the CIFAR-100 dataset, we perform the same data preprocessing techniques as for CIFAR-10 dataset.

## A.3 EXPERIMENTAL SETUP OF OUR PROPOSED METHOD

**CIFAR-10**   For the CIFAR-10 dataset, we use a convolutional SNN structure with 6 convolutional layers and 2 fully-connected layers ([[256C3]×3-MP2]×2-FC×2), similar to the network proposed in previous work (Fang et al., 2021b). Pruning is applied to all neurons and weights in the network, except for neurons in the fully-connected layers. The batch size and simulation step are set to 16 and 8, respectively. We use the Adam optimizer for optimizing both weights $w$ and masks $\alpha_w, \alpha_n$. We train the model for a total of 1000 epochs with an initial learning rate of $\eta = 0.0001$. The subnetwork is produced at epoch 800, and is then fine-tuned for extra 200 epochs.

**DVS-CIFAR10**   For the DVS-CIFAR10 dataset, we use the VGGSNN structure proposed in previous work (Deng et al., 2022), which consists of 8 convolutional layers and one fully-connected layer (64C3-128C3-AP2-[256C3]×2-AP2-[[512C3]×2-AP2]×2-FC). We select the last 1/10 of each category as the test set and the first 9/10 as the training set. The batch size and simulation step are set to 64 and 10, respectively. We train the model for a total of 320 epochs, with 240 epochs dedicated to training-while-pruning and 80 epochs for fine-tuning. We use the SGD optimizer with an initial learning rate of $\eta = 0.025$ and cosine decay to 0. Additionally, we apply data augmentation (Li et al., 2022) and the TET (Deng et al., 2022) loss in all experiments conducted on the DVS-CIFAR10 dataset.

**ImageNet**   For the ImageNet dataset, we use the SEW ResNet18 structure proposed by Fang et al. (2021a). We train the model for a total of 320 epochs, with 280 dedicated to training-while-pruning and 40 epochs for fine-tuning. We follow the setup outlined in Fang et al. (2021a), with a batch size of 256, and a simulation step of 4. We use Adam optimizer with an initial learning rate of $\eta = 0.001$ and cosine decay to 0.

**CIFAR-100**   For the CIFAR-100 dataset, we use the SEW ResNet18 structure (Fang et al., 2021a) with modifications to fit different input sizes. Similar to the other datasets, the model is trained for a total of 320 epochs, with 280 epochs dedicated to pruning and 40 epochs for fine-tuning. The batch size and simulation step are set to 64 and 10, respectively. We use the SGD optimizer with an initial learning rate of $\eta = 0.025$ and cosine decay to 0.

**Hyperparameters**   The key hyperparameters in our proposed method are the weight factor of the penalty term $\lambda$, the initial value of masks $\alpha_0$, the initial and final values of the shape factor of the scaled sigmoid function $\beta_0$ and $\beta_T$. We set $\alpha_0 = 0$, $\beta_0 = 5$, and $\beta_T = 1000$ in ALL experiments, and adjust the sparsity of the network by varying the value of $\lambda$.

## A.4 DETAILED EXPERIMENTAL RESULTS

Tab. A1 lists the experimental results under more sparsity levels, as well as the experimental results on the CIFAR-100 dataset. For CIFAR10 dataset, we add results of $\lambda = 1 \times 10^{-11}$ and $\lambda = 1 \times 10^{-10}$. For the DVS-CIFAR10 dataset, we add the result of $\lambda = 5 \times 10^{-10}$. We also add experimental results on the CIFAR-100 dataset with $\lambda$ varies from $2 \times 10^{-10}$ to $3 \times 10^{-9}$. On both the CIFAR10, DVS-CIFAR10, and CIFAR100 dataset, our proposed method can effectively reduce the average SOPs to less than 10M, with an accuracy loss of at most 4.1%. This demonstrates that our method can effectively reduce energy consumption.

The detailed results of the ablation study are listed in Tab. A2. Comparing with the results on the CIFAR10 dataset in Tab. A1, our proposed method achieves better performance than both the pruning weight-only method and the pruning neuron-only method.

## A.5 EXPERIMENTAL SETUP OF COMPARISON EXPERIMENTS

Tab. A3 provides the general experimental settings for the comparison experiments. Extra hyperparameters for each method can be found in Tab. A4.

Note that all the compared methods employ different approaches to control sparsity. For ADMM (Deng et al., 2021), we conduct separate experiments with target sparsity levels set to 0.8, 0.9, and 0.95. For GradR (Chen et al., 2021), we set the penalty $\alpha$ to $2 \times 10^{-4}$, $5 \times 10^{-4}$, $1 \times 10^{-3}$,

Table A1: Detailed experimental results

| Dataset | Arch. | $\lambda$ | Top-1 Acc. (%) | Acc. Loss (%) | Avg. SOPs (M) | Ratio | Conn. (%) | Neuron (%) | Weight (%) |
|---|---|---|---|---|---|---|---|---|---|
| CIFAR10 | 6 Conv, 2 FC | (dense) | 92.84 | 0 | 776.12 | 1 | 100 | 100 | 100 |
| | | $5 \times 10^{-12}$ | 92.63 | -0.21 | 38.32 | 20.25 | 2.97 | 21.03 | 34.24 |
| | | $1 \times 10^{-11}$ | 92.28 | -0.56 | 25.81 | 30.07 | 1.91 | 16.27 | 30.41 |
| | | $2 \times 10^{-11}$ | 92.05 | -0.79 | 16.47 | 47.12 | 1.16 | 12.27 | 26.05 |
| | | $3 \times 10^{-11}$ | 91.47 | -1.37 | 11.98 | 64.78 | 0.87 | 10.31 | 23.00 |
| | | $5 \times 10^{-11}$ | 90.65 | -2.19 | 8.50 | 91.31 | 0.63 | 8.23 | 19.34 |
| | | $1 \times 10^{-10}$ | 89.37 | -3.47 | 5.03 | 154.30 | 0.38 | 5.73 | 13.65 |
| DVS-CIFAR10 | VGGSNN | (dense) | 82.4 | 0 | 647.30 | 1 | 100 | 100 | 100 |
| | | $1 \times 10^{-10}$ | 81.9 | -0.5 | 47.81 | 13.54 | 6.80 | 42.73 | 36.35 |
| | | $2 \times 10^{-10}$ | 81.0 | -1.4 | 31.86 | 20.32 | 4.46 | 35.13 | 33.02 |
| | | $5 \times 10^{-10}$ | 81.0 | -1.4 | 14.66 | 44.15 | 1.95 | 23.05 | 25.88 |
| | | $7 \times 10^{-10}$ | 79.0 | -3.4 | 10.02 | 64.60 | 1.27 | 18.56 | 22.56 |
| | | $1 \times 10^{-9}$ | 78.3 | -4.1 | 6.75 | 95.90 | 0.77 | 14.11 | 18.66 |
| CIFAR100 | SEW ResNet18 | (dense) | 74.16 | 0 | 250.91 | 1 | 100 | 100 | 100 |
| | | $2 \times 10^{-10}$ | 72.34 | -1.82 | 27.16 | 9.24 | 11.76 | 62.07 | 29.35 |
| | | $3 \times 10^{-10}$ | 71.99 | -2.17 | 24.53 | 10.23 | 10.36 | 56.56 | 27.98 |
| | | $5 \times 10^{-10}$ | 71.43 | -2.73 | 21.16 | 11.86 | 8.67 | 51.59 | 25.93 |
| | | $1 \times 10^{-9}$ | 71.31 | -2.85 | 15.54 | 16.15 | 6.10 | 40.97 | 21.96 |
| | | $2 \times 10^{-9}$ | 70.45 | -3.71 | 9.60 | 26.14 | 3.60 | 28.74 | 17.07 |
| | | $3 \times 10^{-9}$ | 69.41 | -4.75 | 6.79 | 36.95 | 2.48 | 21.89 | 14.38 |
| ImageNet | SEW ResNet18 | (dense) | 63.18 | 0 | 1337.87 | 1 | 100 | 100 | 100 |
| | | $5 \times 10^{-11}$ | 61.89 | -1.29 | 311.70 | 4.29 | 28.60 | 74.86 | 35.26 |
| | | $1 \times 10^{-10}$ | 60.00 | -3.18 | 177.99 | 7.52 | 19.31 | 68.27 | 27.74 |
| | | $2 \times 10^{-10}$ | 58.99 | -4.19 | 116.88 | 11.45 | 14.19 | 51.17 | 23.52 |

Table A2: Experimental results of ablation study

| Method | $\lambda$ | Top-1 Acc. (%) | Acc. Loss (%) | Avg. SOPs (M) | Conn. (%) | Weight (%) | Neuron (%) |
|---|---|---|---|---|---|---|---|
| Pruning Weight Only | $1 \times 10^{-10}$ | 92.10 | -0.74 | 24.90 | 1.35 | 17.41 | 100 |
| | $2 \times 10^{-10}$ | 91.85 | -0.99 | 18.40 | 0.94 | 12.15 | 100 |
| | $3 \times 10^{-10}$ | 91.41 | -1.43 | 15.51 | 0.74 | 9.36 | 100 |
| | $5 \times 10^{-10}$ | 91.04 | -1.80 | 11.86 | 0.55 | 6.47 | 100 |
| | $1 \times 10^{-9}$ | 89.82 | -3.02 | 8.62 | 0.30 | 3.85 | 100 |
| Pruning Neuron Only | $1 \times 10^{-10}$ | 91.49 | -1.35 | 27.29 | 2.88 | 100 | 4.51 |
| | $2 \times 10^{-10}$ | 90.71 | -2.13 | 20.47 | 2.21 | 100 | 3.39 |
| | $3 \times 10^{-10}$ | 89.96 | -2.88 | 16.54 | 1.83 | 100 | 2.75 |

Table A3: General experimental settings

| Dataset | Method | Epochs | Optimizer | Learning Rate | Batch Size | Time Steps |
|---|---|---|---|---|---|---|
| CIFAR10 | ADMM | 235 | Adam | 0.001 | 50 | 8 |
| | GradR | 2048 | Adam | 0.0001 | 16 | 8 |
| | ESLSNN | 500 | Adam | 0.001[1] | 64 | 2 |
| | STDS | 2048 | Adam | 0.0001 | 16 | 8 |
| | Ours | 1000 | Adam | 0.0001 | 16 | 8 |
| DVS-CIFAR10 | ESLSNN | 320 | SGD | 0.1[1] | 64 | 10 |
| | STDS | 320 | SGD | 0.025[1] | 64 | 10 |
| | Ours | 320 | SGD | 0.025[1] | 64 | 10 |
| ImageNet | STDS | 320 | SGD | 0.1[1] | 256 | 4 |
| | Ours | 320 | Adam | 0.001[1] | 256 | 4 |
| CIFAR100 | Ours | 320 | SGD | 0.025[1] | 64 | 4 |

[1] Cosine decay

$2 \times 10^{-3}$, $5 \times 10^{-3}$, the same as the original settings. Similarly, for STDS (Chen et al., 2022), we set the target threshold $D$ to 1.0, 2.0, 3.0, 4.0, and 5.0 on CIFAR10 dataset and 0.1, 0.6, 0.8, 1.5, 3.0 and 5.0, on ImageNet dataset, the same as the original settings. For the experiments of STDS

Table A4: Extra experimental settings

| Method | Description | Value |
|---|---|---|
| ADMM | Pre-training epochs | $200^1$ |
| | ADMM re-training epochs | 20 |
| | Hard pruning re-training epochs | 15 |
| | Penalty coefficient for ADMM | $5 \times 10^{-4}$ |
| GradR | Target sparsity | 0.95 |
| STDS | Threshold scheduler | Sine |
| ESLSNN | Pruning rule | Magnitude[2] and SET[3] |
| | Growth rule | Momentum |
| Ours | Initial value of masks | 0 |
| | Initial value of scale factor | 5 |
| | Final value of scale factor | 1000 |

[1] The original setting is 100, we extend the pre-training epochs to improve performance
[2] For CIFAR10
[3] For DVS-CIFAR10

on DVS-CIFAR10 in our implementation, we set $D$ to 0.05, 0.2, and 0.5. For ESLSNN (Shen et al., 2023), we set the target sparsity to 0.8, 0.9, 0.95.

## A.6 VISUALIZATION

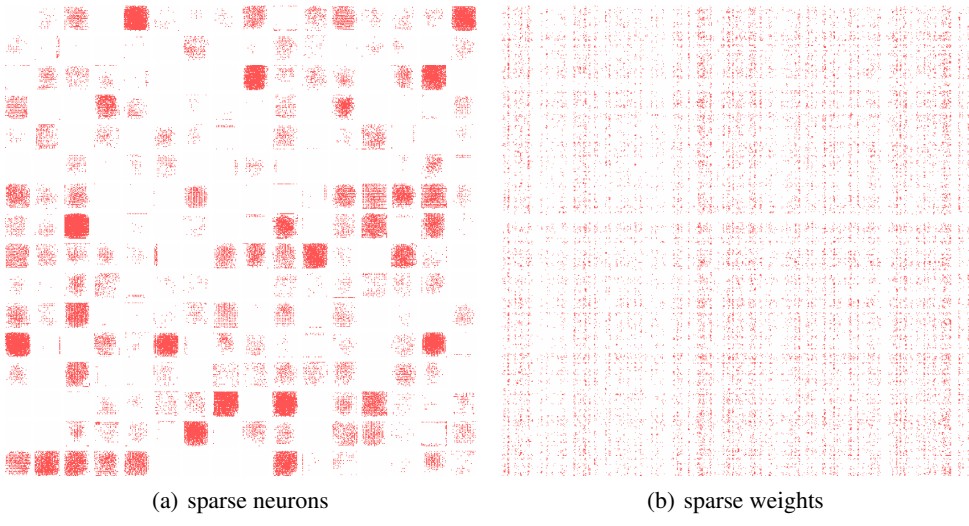

(a) sparse neurons          (b) sparse weights

Figure A1: Visualization of sparse neurons and weights

In this section, we visualize the sparse structure obtained by pruning the network using our method. Fig. A1(a) illustrates the sparsity pattern of the encoding layer neurons in the sparse CifarNet (the 6 Conv, 2 FC network used on the CIFAR10 dataset; we refer to this network as CifarNet in the following) with $\lambda = 1 \times 10^{-11}$. In the figure, each small square represents a $32 \times 32$ feature map, and the color of each pixel represents the state of a neuron. Red indicates an active (non-pruned) neuron, while white indicates a pruned neuron. Fig. A1(b) illustrates the sparsity pattern of the weights in the first convolutional layer of the same sparse CifarNet. Each row corresponds to an input channel, and each column corresponds to an output channel. The $3 \times 3$ little square at each intersection corresponds to a convolution kernel.

From Fig. A1(a), we observe that the neurons in the encoding layer form a self-organizing multi-resolution structure. Some channels are nearly dense, with only a few neurons pruned. These channels have the full resolution and retain the most information. Some channels have a moderate level of sparsity, with around 50% to 25% of neurons remaining, resulting in reduced resolution. In some channels, only a few neurons remain, capturing higher-level information at lower resolutions. Redundant channels have been completely removed.

Understanding the sparse structure of the convolutional kernels is challenging. However, we can observe that the convolutional kernels corresponding to the completely pruned channels are also fully pruned.

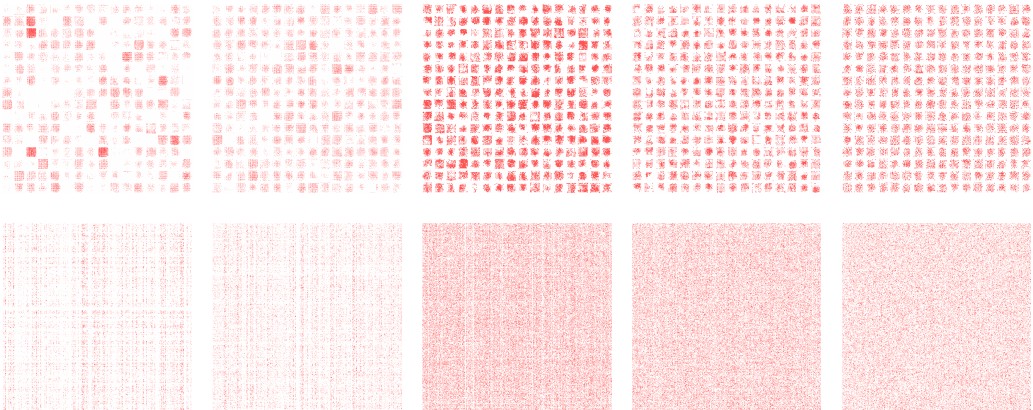

Figure A2: Visualization of sparse neurons and weights in each convolution layer

Fig. A2 visualize the sparse structure of the other layers. From left to right, the first to fifth convolutional layers are shown in order. It can be seen that the difference in sparsity between channels tends to be smaller for the deeper layers. This is because, for deeper layers, the output represents higher-level feature maps that no longer exhibit a multi-resolution structure. Consequently, the sparsity patterns become more homogeneous across channels.

## A.7 HYPERPARAMETER ANALYSIS

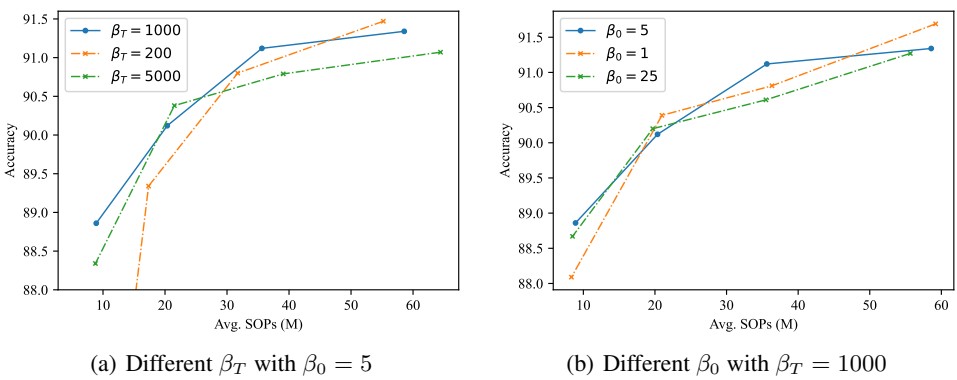

(a) Different $\beta_T$ with $\beta_0 = 5$      (b) Different $\beta_0$ with $\beta_T = 1000$

Figure A3: Comparison of performance under different $\beta_T$ and $\beta_0$

In the main text, we set $\beta_0 = 5$ and $\beta_T = 1000$ providing without further explanation. In this section, we investigate how these hyperparameters, $\beta_0$ and $\beta_T$, affect the performance of our proposed method.

Our experimental setup is as follows. We train CifarNet on the CIFAR10 dataset for a total of 150 epochs, with 100 epochs dedicated to pruning and 50 epochs for fine-tuning. We vary one

Table A5: Performance comparison between the proposed method and other pruning methods on ImageNet dataset

| Pruning Method | Arch. | T | Top-1 Acc. (%) | Acc. Loss (%) | Avg. SOPs (M) | Ratio | Conn. (%) | Param. (M) |
|---|---|---|---|---|---|---|---|---|
| ADMM | SEW ResNet18 | 4 | 59.48 | -3.74 | 402.75 | 3.33 | 29.25 | 1.94 |
| | | | 55.85 | -7.37 | 262.39 | 5.11 | 21.04 | 1.24 |
| | | | 50.57 | -12.65 | 173.10 | 7.76 | 15.68 | 0.75 |
| STDS | SEW ResNet18 | 4 | 62.84 | -0.34 | 1082.60 | 1.24 | 77.54 | 7.42 |
| | | | 61.51 | -1.67 | 512.95 | 2.61 | 34.79 | 2.38 |
| | | | 61.30 | -1.88 | 424.07 | 3.15 | 29.29 | 1.94 |
| | | | 59.93 | -3.25 | 273.39 | 4.89 | 21.08 | 1.24 |
| | | | 58.06 | -5.12 | 182.49 | 7.33 | 15.71 | 0.75 |
| | | | 56.28 | -6.90 | 134.64 | 9.94 | 12.95 | 0.52 |
| **This work** | SEW ResNet18 | 4 | **61.89** | **-1.29** | **311.70** | **4.29** | **28.60** | **3.93** |
| | | | **60.00** | **-3.18** | **177.99** | **7.52** | **19.31** | **3.10** |
| | | | **58.99** | **-4.19** | **116.88** | **11.45** | **14.19** | **2.62** |

hyperparameter while keeping the other fixed. Specifically, we set $\lambda$ to $5 \times 10^{-12}$, $1 \times 10^{-11}$, $2 \times 10^{-11}$, and $5 \times 10^{-11}$, respectively, to assess the performance under different sparsity levels. Other settings are inherited from the experiments in the main text and have been detailed above.

We first analyze the effect of different values of $\beta_T$ on performance. As shown in Fig. 3(a), we observe that a smaller value of $\beta_T$, such as $\beta_T = 200$, yields slightly better performance under low sparsity conditions ($\lambda = 5 \times 10^{-12}$). However, it leads to significant degradation under high sparsity conditions ($\lambda = 5 \times 10^{-11}$, 5.70M SOPs with 81.64% accuracy, not shown in the figure). This degradation occurs because a too-small $\beta_T$ prevents the scaled sigmoid function from adequately approximating the Heaviside step function. Consequently, there is a substantial gap between pruning and fine-tuning due to the replacement of the not fully converged continuous masks with binary masks. On the other hand, $\beta_T = 5000$ achieves equal or slightly better performance under high sparsity conditions ($\lambda = 5 \times 10^{-11}$ and $\lambda = 2 \times 10^{-11}$), but slightly impairs performance under low sparsity conditions ($\lambda = 5 \times 10^{-12}$ and $\lambda = 1 \times 10^{-11}$).

Next, we examine the impact of $\beta_0$ on performance. In fact, $\beta_0$ has less impact on performance. As shown in Fig. 3(b), we observe that $\beta_0 = 1$, $\beta_0 = 5$, and $\beta_0 = 25$ achieve nearly identical performance when $\lambda = 5 \times 10^{-12}$ or $2 \times 10^{-11}$. Similar to the case of $\beta_T$, using a small value of $\beta_0$ leads to a slight degradation under high sparsity conditions. It remains unclear why there is a slight degradation for $\beta_0 = 1$ and 25 when $\lambda = 1 \times 10^{-11}$.

In summary, the key point is to avoid using excessively small values of $\beta_T$ for high sparsity levels. Furthermore, $\beta_0$ exhibits greater robustness compared to $\beta_T$, indicating that careful adjustment of $\beta_0$ is unnecessary.

## A.8 COMPARISON ON IMAGENET DATASET

Tab. A5 lists a detailed comparison on ImageNet dataset between the proposed method and other pruning methods including ADMM and STDS. Experimental results show that our method achieves comparable performance while significantly reducing energy consumption compared to STDS and ADMM.

## A.9 DIFFERENCE BETWEEN ANNS AND SNNS IN UNSTRUCTURED NEURON PRUNING

The reason why unstructured neuron pruning is more suitable for SNNs lies in the sparse computing manner of SNNs and neuromorphic hardware. In ANNs, neurons primarily function as stateless activation functions, such as Sigmoid or ReLU. These activation functions typically yield limited output sparsity. Consequently, ANNs heavily rely on dense matrix computations and are commonly implemented on hardware architectures optimized for dense computing, such as GPUs. These architectures are not designed to effectively exploit the unstructured sparsity of neurons. In contrast, SNNs employ sparse event-driven computing as their neurons communicate with each other by fir-

ing sparse, discrete spikes. Thus, SNNs are typically implemented on neuromorphic hardware that is highly optimized for sparsity. Therefore, unstructured neuron pruning is more suitable for SNNs.

It is important to note that the above analysis is only for the common case. In fact, some work has successfully utilized sparsity in ANNs by designing novel hardware (Han et al., 2016), utilizing AVX instructions on CPUs (Kurtz et al., 2020), or designing new algorithms for GPUs (Wang, 2020). We highly appreciate these efforts. However, it is crucial to note that these methods of exploiting sparsity are not widely adopted yet. Therefore, the analysis presented above is generally reasonable as it applies to most cases.

## A.10 Detailed Discussion on Hardware Suitability of Our Energy Model

In the following, we discuss in detail which architectures are suited to use SOPs as an energy consumption metric and which are not.

First, we would like to emphasize the significance of SOPs as an important metric in both the hardware and software domains. Many works on neuromorphic hardware provide the energy consumption per SOP (or equally, energy per synaptic spike op, energy per connection, SOPs per second per Watt (SOPS/W), etc.) as an important metric for assessing hardware energy efficiency. Notable examples include Loihi (Davies et al., 2018) (23.6pJ per synaptic spike op), TrueNorth (Merolla et al., 2014) (46 GSOPS/W (typical), 400 GSOPS/W (max)), and TianJic (Pei et al., 2019) (649 GSOPS/W (max)). As stated in (Furber, 2016), "Since synaptic processing dominates the system energy cost, this (denotes energy per (synaptic) connection) is the best guide to the overall energy efficiency of the system." For this reason, many SNN works use SOPs as the metric for estimating their energy consumption estimation, such as ESLSNN (Shen et al., 2023) compared in our manuscript. Therefore, it is reasonable to use SOPs as a meaningful energy consumption metric.

While SOPs is an important metric, we recognize that it does not cover all aspects of hardware energy consumption like memory access and data movement. Therefore, for architectures where other aspects of energy consumption, such as memory accesses, dominate and are not proportional to SOPs, it may not be accurate to use SOPs as the only metric of energy consumption. We list two types of these architectures. The first is the memory access-dominated architectures, such as SATA (Yin et al., 2023). As demonstrated in (Bhattacharjee et al., 2023), the energy consumption associated with memory access in SATA is significantly higher than that of computing and significantly depends on the dataflow. The second is the analog-based architectures such as (Qiao et al., 2015). Architectures with analog synapses typically consume much more energy to generate a spike compared to synaptic computation, thus their energy consumption may not be proportional to SOPs.

However, we believe that there are architectures that match our energy model due to their full exploitation of sparsity. By fully utilizing sparsity, the energy consumption of the whole system may be reduced to a level that is linear to the sparsity of synaptic operations, if the other aspects of energy consumption are also sparsely triggered by the synaptic operations. Loihi (Davies et al., 2018) is exactly such kind of architecture. Loihi is a highly optimized neuromorphic chip designed for efficient neuromorphic computing. It incorporates two key features to enhance energy efficiency: local memory access and asynchronous spike communication. By colocating memory and computing within the core, Loihi minimizes the need for global memory access and reduces data travel distances. Although the detailed architecture design of Loihi has not been made public, the Loihi team claims that Loihi's neuromorphic mesh and cores are built with asynchronous circuits and, at the transistor level, communicate event-driven tokens of information between logic stages, which helps utilize the sparsity (Davies et al., 2021). While there is no direct evidence specific to SNNs, experimental results on other tasks indicate that the energy consumption of Loihi is roughly proportional to the computing. We believe that a high degree of optimization that exploits sparsity in memory access, data communication, and computation is a key point in making the overall energy consumption roughly linear with the amount of computation.

