# OpenReview forum: "Towards Energy Efficient Spiking Neural Networks: An Unstructured Pruning Framework"
_ICLR.cc/2024/Conference — ICLR 2024 spotlight_

### Official Review · Reviewer_tCqk · 2023-10-26

**Soundness:** 3 good
**Presentation:** 3 good
**Contribution:** 4 excellent
**Rating:** 8
**Confidence:** 5

**Summary:**

Low energy consumption is a main property of spiking neural networks. This paper focuses on enhancing energy efficiency of spiking neural networks. The authors first propose the energy consumption model of SNNs and then try to optimize the energy efficiency by introducing the unstructured weight and neuron pruning framework with masks. Besides, a new energy penalty term is proposed to solve the ill-posed problem. Experimental results on CIFAR10, DVS-CIFAR10 and ImageNet datasets demonstrate the proposed method can achieve the SOTA energy efficiency with comparable performance.

**Strengths:**

The paper is the first application of unstructured neuron pruning to deep SNNs. The combination of unstructured weight pruning and unstructured neuron pruning is interesting and efficient. Besides, the authors propose some insight methods to address the ill-posed problem when jointly pruning neurons and weights under energy constraints.

**Weaknesses:**

1. The difference between ANNs and SNNs in pruning (especially unstructured neuron pruning) is not fully explained. It is not clear why unstructured neuron pruning is not used in ANNs. Why is it more suitable for SNNs?
2. The authors mainly illustrate how to optimize the penalty, but there was no explanation of how to optimize the first term of the loss function in Eq.11.

**Questions:**

Can the authors explain why unstructured neuron pruning is not (commonly) used in ANNs?  How to optimize the first term of the loss function in Eq.11? How to choose the parameter $\beta$ for different datasets?

---

> ### Author Response · Authors · 2023-11-15
> **Rebuttal by Authors**
>
> We appreciate the time and effort you invested in reviewing our paper. We will address all the issues you have raised.
>
> > The difference between ANNs and SNNs in pruning (especially unstructured neuron pruning) is not fully explained. It is not clear why unstructured neuron pruning is not used in ANNs. Why is it more suitable for SNNs?
>
> Thanks for pointing it out! The reason why unstructured neuron pruning is more suitable for SNNs lies in the sparse computing manner of SNNs and neuromorphic hardware. In ANNs, neurons primarily function as stateless activation functions, such as Sigmoid or ReLU. These activation functions typically yield limited output sparsity. Consequently, ANNs heavily rely on dense matrix computations and are commonly implemented on hardware architectures optimized for dense computing, such as GPUs. These architectures are not designed to effectively exploit the unstructured sparsity of neurons. In contrast, SNNs employ sparse event-driven computing as their neurons communicate with each other by firing sparse, discrete spikes. Thus, SNNs are typically implemented on neuromorphic hardware that is highly optimized for sparsity. Therefore, unstructured neuron pruning is more suitable for SNNs.
>
> It is important to note that the above analysis is only for the common case. In fact, some work has successfully utilized sparsity in ANNs by designing novel hardware [1], utilizing AVX instructions on CPUs [2], or designing new algorithms for GPUs [3]. We highly appreciate these efforts. However, it is crucial to note that these methods of exploiting sparsity are not widely adopted yet. Therefore, the analysis presented above is generally reasonable as it applies to most cases.
>
> We have added this detailed analysis in the appendix.
>
>
> > The authors mainly illustrate how to optimize the penalty, but there was no explanation of how to optimize the first term of the loss function in Eq.11.
>
> If we understand correctly, the first term of the loss function here refers to $\sigma(\beta{\boldsymbol \alpha}_w)\odot{\boldsymbol w}$, and you are asking why the element-wise multiplication form in this term isn't improved as the penalty term does. In fact, the form of this term does not require further improvement, since the element-wise multiplication here is not problematic.
>
> This is because the loss function term is different from the penalty term. The original form of the penalty term in Eq. 8 is a simple summation of the products of the three types of elements. For each of these products, if one of the three elements takes 0, the product achieves the minimum value of 0. Therefore, this penalty term tends to prune only one type of element, making it an ill-posed problem that is very sensitive to initial conditions. In contrast, the loss function term measures the loss between the target and the model output. From the perspective of the term $\sigma(\beta{\boldsymbol \alpha}_w)\odot{\boldsymbol w}$, the loss function term is a complex nonlinear term with respect to $\sigma(\beta{\boldsymbol \alpha}_w)\odot{\boldsymbol w}$, thus there is not a trivial solution where $\boldsymbol \alpha_w$ or $\boldsymbol w$ being 0 would make the loss function term achieve the minimum value of 0. Thus using element-wise multiplication here is not problematic.
>
> For how to solve the optimization problem in Eq. 11, please refer to the pseudo-code using the stochastic gradient descent (SGD) method in Section A.1 on page 14.
>
> > How to choose the parameter $\beta$ for different datasets?
>
> We would like to clarify that there is no necessity to choose different values of $\beta$ for different datasets, as the parameter $\beta$ is robust across different datasets. We consistently utilize the settings of $\beta_0=5$ and $\beta_T=1000$ in all of our experiments conducted on different datasets. Experimental results show that our parameter setting achieves good performance across diverse datasets, demonstrating the robustness of parameter $\beta$. Moreover, this robustness facilitates the application of our method to different tasks without the need for additional hyperparameter searches.
>
>
>
>
> [1] Han, S., Liu, X., Mao, H., Pu, J., Pedram, A., Horowitz, M. A., & Dally, W. J. (2016). EIE: Efficient inference engine on compressed deep neural network. *ACM SIGARCH Computer Architecture News*, *44*(3), 243-254.
>
> [2] Kurtz, M., Kopinsky, J., Gelashvili, R., Matveev, A., Carr, J., Goin, M., ... & Alistarh, D. (2020, November). Inducing and exploiting activation sparsity for fast inference on deep neural networks. In *International Conference on Machine Learning* (pp. 5533-5543). PMLR.
>
> [3] Wang, Z. (2020, September). SparseRT: Accelerating unstructured sparsity on GPUs for deep learning inference. In *Proceedings of the ACM international conference on parallel architectures and compilation techniques* (pp. 31-42).

---

> ### Comment · Reviewer_tCqk · 2023-11-23
> **Thanks for the clarification**
>
> Great to see the difference between ANNs and SNNs in unstructured neuron pruning, which is helpful to better assess the contributions of the paper. It was a pleasure to read the manuscript, and I am happy to increase my rating.

---

### Official Review · Reviewer_BChF · 2023-10-28

**Soundness:** 2 fair
**Presentation:** 3 good
**Contribution:** 2 fair
**Rating:** 3
**Confidence:** 3

**Summary:**

This paper describes a means of improving the energy consumption of SNNs via irregular pruning, that includes pruning of nodes as well as weights. The reduction in number of synaptic operations is significant with only small drops in accuracy.  The method involves adjusting the loss function with several reasonable approximations.

**Strengths:**

The reports results are strong and the paper is very well written with clear explanations of the approach taken.  The ability to target a given sparsity level is a plus.

**Weaknesses:**

The cost function associated with number of synaptic operations ignores the cost associated with managing the sparsity. In CNNs it is well known that it is hard to take advantage of irregular non-structured sparsity (see e.g., https://arxiv.org/pdf/1907.02124.pdf) and this is somewhat well understand in SNNs as well. Many hardware accelerators claim to try to take advantage of sparsity, but the cost associated with managing the sparsity reduces the overall energy benefits.  See https://arxiv.org/pdf/2309.03388.pdf for an example of a good paper that identifies a high-level energy model of SNNs which captures this issue by more accurately capturing the associated memory cost of managing an SNN.

The proposed approach ignores these issues and only uses SOPs as a metric of energy consumption. At the minimum, I would argue that the paper needs to appreciate that this metric is not accurate for many hardware accelerators and highlight which architectures this metric is most suited. I hypothesize, for example, SOPs is more reasonable for many core processors like Loihi (which do not focus on weight re-use) and less accurate for architectures where dataflow such as weight stationary play a key role.

**Questions:**

1. Which accelerator architectures match well to using SOP as a metric and which do not?

---

> ### Author Response · Authors · 2023-11-15
> **Rebuttal by Authors Part (1/2)**
>
> Thank you for your valuable feedback on our work. Your insights on accelerator architectures are greatly appreciated and will help us to improve our paper. We agree with you that our model may not be suitable for all architectures, since many architectures claim that they take advantage of sparsity but have not fully exploited it, leading to additional costs for managing the sparsity. In the following, we discuss in detail which architectures are suited to use SOPs as an energy consumption metric and which are not.
>
> First, we would like to emphasize the significance of SOPs as an important metric in both the hardware and software domains. Many works on neuromorphic hardware provide the energy consumption per SOP (or equally, energy per synaptic spike op, energy per connection, SOPs per second per Watt (SOPS/W), etc.) as an important metric for assessing hardware energy efficiency. Notable examples include Loihi [1] (23.6pJ per synaptic spike op), TrueNorth [2] (46 GSOPS/W (typical), 400 GSOPS/W (max)), and TianJic [3] (649 GSOPS/W (max)). As stated in [4], "Since synaptic processing dominates the system energy cost, this (denotes energy per (synaptic) connection) is the best guide to the overall energy efficiency of the system." For this reason, many SNN works use SOPs as the metric for estimating their energy consumption estimation, such as ESLSNN [5] compared in our manuscript. Therefore, it is reasonable to use SOPs as a meaningful energy consumption metric.
>
> While SOPs is an important metric, we recognize that it does not cover all aspects of hardware energy consumption like memory access and data movement. Therefore, for architectures where other aspects of energy consumption, such as memory accesses, dominate and are not proportional to SOPs, it may not be accurate to use SOPs as the only metric of energy consumption. We list two types of these architectures. The first is the memory access-dominated architectures, such as SATA [6]. As demonstrated in [7], the energy consumption associated with memory access in SATA is significantly higher than that of computing and significantly depends on the dataflow. The second is the analog-based architectures such as [8]. Architectures with analog synapses typically consume much more energy to generate a spike compared to synaptic computation, thus their energy consumption may not be proportional to SOPs.
>
> However, we believe that there are architectures that match our energy model due to their full exploitation of sparsity. By fully utilizing sparsity, the energy consumption of the whole system may be reduced to a level that is linear to the sparsity of synaptic operations, if the other aspects of energy consumption are also sparsely triggered by the synaptic operations. However, due to the wide variety of hardware architectures, it is almost impossible to distinguish which architecture can fully exploit sparsity by summarizing certain characteristics like many-core, weight reuse, the role of dataflow, etc. Thus, we list two different architectures that we believe may match our energy model and discuss which kind of architectures may match our energy model by analyzing their characteristics.
>
> 1. Loihi [1]. Loihi is a highly optimized neuromorphic chip designed for efficient neuromorphic computing. It incorporates two key features to enhance energy efficiency: local memory access and asynchronous spike communication. By colocating memory and computing within the core, Loihi minimizes the need for global memory access and reduces data travel distances. Although the detailed architecture design of Loihi has not been made public, the Loihi team claims that Loihi’s neuromorphic mesh and cores are built with asynchronous circuits and, at the transistor level, communicate event-driven tokens of information between logic stages, which helps utilize the sparsity [9]. While there is no direct evidence specific to SNNs, experimental results on other tasks indicate that the energy consumption of Loihi is roughly proportional to the computing.
> 2. SpinalFlow [10]: SpinalFlow differs significantly from Loihi. It is based on the DCNN accelerator Eyeriss [11], making its overall architecture more similar to an ANN accelerator rather than neuromorphic hardware. It doesn't have local memory access and asynchronous communication features like Loihi. Instead, it employs a global buffer and is built with the synchronous circuit. SpinalFlow employs an SNN-tailored dataflow and increases the reuse of neuron potential, input spikes, and weights. The architecture is designed to effectively leverage the inherent sparsity in spike-based computations. Experimental results demonstrate that SpinalFlow successfully takes advantage of sparsity and reduces energy consumption to a level proportional to sparsity.

---

> > ### Author Response · Authors · 2023-11-15
> > **Rebuttal by Authors Part (2/2)**
> >
> > The aforementioned architectures, Loihi and SpinalFlow, differ significantly in their design perspectives, with Loihi from a neuromorphic computing perspective and SpinalFlow from an AI accelerator perspective. Consequently, certain characteristics of these architectures, such as asynchronous/synchronous circuits and local/global memory access, are diametrically opposed. However, both architectures have been highly optimized to utilize the sparsity in memory access, data communication, and computation. We believe this is the key point that makes the overall energy consumption roughly linear to computing.
> >
> > We have revised our manuscript to point out that our energy model is not suitable for all architectures and highlight that it matches the architectures that are highly optimized for sparsity in the main text. We have added this detailed discussion to the appendix.
> >
> >
> >
> >
> >
> > [1] Davies, M., Srinivasa, N., Lin, T.-H., Chinya, G., Cao, Y., Choday, S. H., Dimou, G., Joshi, P., Imam, N., & Jain, S. (2018). Loihi: A neuromorphic manycore processor with on-chip learning. *IEEE Micro*, 38(1), 82-99.
> >
> > [2] Merolla, P. A., Arthur, J. V., Alvarez-Icaza, R., Cassidy, A. S., Sawada, J., Akopyan, F., Jackson, B. L., Imam, N., Guo, C., & Nakamura, Y. (2014). A million spiking-neuron integrated circuit with a scalable communication network and interface. *Science*, 345(6197), 668-673.
> >
> > [3] Pei, J., Deng, L., Song, S., Zhao, M., Zhang, Y., Wu, S., Wang, G., Zou, Z., Wu, Z., & He, W. (2019). Towards artificial general intelligence with hybrid Tianjic chip architecture. *Nature*, 572(7767), 106-111.
> >
> > [4] Furber, S. (2016). Large-scale neuromorphic computing systems. *Journal of neural engineering*, 13(5), 051001.
> >
> > [5] Shen, J., Xu, Q., Liu, J. K., Wang, Y., Pan, G., & Tang, H. (2023). ESL-SNNs: An Evolutionary Structure Learning Strategy For Spiking Neural Networks. *Proceedings of the AAAI Conference on Artificial Intelligence*.
> >
> > [6] Yin, R., Moitra, A., Bhattacharjee, A., Kim, Y., & Panda, P. (2022). SATA: Sparsity-aware training accelerator for spiking neural networks. *IEEE Transactions on Computer-Aided Design of Integrated Circuits and Systems*.
> >
> > [7] Bhattacharjee, A., Yin, R., Moitra, A., & Panda, P. (2023). Are SNNs Truly Energy-efficient? $-$ A Hardware Perspective. *arXiv preprint arXiv:2309.03388*.
> >
> > [8] Qiao, N., Mostafa, H., Corradi, F., Osswald, M., Stefanini, F., Sumislawska, D., & Indiveri, G. (2015). A reconfigurable on-line learning spiking neuromorphic processor comprising 256 neurons and 128K synapses. *Frontiers in neuroscience*, *9*, 141.
> >
> > [9] Davies, M., Wild, A., Orchard, G., Sandamirskaya, Y., Guerra, G. A. F., Joshi, P., Plank, P., & Risbud, S. R. (2021). Advancing neuromorphic computing with loihi: A survey of results and outlook. Proceedings of the IEEE, 109(5), 911-934.
> >
> > [10] Narayanan, S., Taht, K., Balasubramonian, R., Giacomin, E., & Gaillardon, P. E. (2020, May). SpinalFlow: An architecture and dataflow tailored for spiking neural networks. In 2020 ACM/IEEE 47th Annual International Symposium on Computer Architecture (ISCA) (pp. 349-362). IEEE.
> >
> > [11] Chen, Y. H., Krishna, T., Emer, J. S., & Sze, V. (2016). Eyeriss: An energy-efficient reconfigurable accelerator for deep convolutional neural networks. *IEEE journal of solid-state circuits*, 52(1), 127-138.

---

> > ### Author Response · Authors · 2023-11-22
> > **Thank you for your time and look forward to your feedback**
> >
> > Dear Reviewer BChF,
> >
> > We have re-evaluated the applicability of our proposed energy consumption model to different hardware architectures in our response. While we emphasize SOP as an important metric, we also recognize its limitations. We have listed several types of architectures for which the metric is inaccurate, as well as architectures for which the metric is suited. We analyze in detail the characteristics of these architectures and the reasons why our energy model is or is not applicable to them. We believe that our energy model is suitable for the architectures that are highly optimized to exploit the sparsity in memory access, data communication, and computation.
> >
> > We have revised our manuscript to point out that our energy model is not suitable for all architectures and highlight that it matches the architectures that are highly optimized for sparsity in the main text. We have added the detailed discussion in our response to the appendix.
> >
> > Based on these facts and positive feedback from other reviewers, we sincerely hope you can reconsider your initial rating. If you still have any further comments or questions, please let us know and we will be glad to address your further concerns.

---

> > ### Comment · Reviewer_BChF · 2023-11-22
> > **Comparison to SpinalFlow not clear**
> >
> > I appreciate the response to my question about energy models and application to various SNN accelerators. And, I completely understand and agree that the model and applicability to Loihi and its variants is valid.
> >
> > However, as I understand SpinalFlow, it is suitable for a temporal coded SNNs. My understanding is SOTA SNN architectures have deviated away from such SNNs (towards direct coded SNNs) as they require too many time steps and thus in the end consume too much energy.   Are your SNNs temporally coded such that application to SpinalFlow makes sense?  I am also unclear how SpinalFlow will adjust to your neuronal pruning as I appreciate it's focus is on the sparsity of the input spike train, not the sparsity of the neurons. Can you clarify?

---

> > > ### Author Response · Authors · 2023-11-23
> > > **Thanks for your response**
> > >
> > > Thank you for pointing this out! We recognize that SpinalFlow is suitable for temporal coded SNNs and focuses on the sparsity of the input spike train, whereas our SNNs are direct coded (i.e., coded by the first layer) and our method focuses on the sparsity of the connections. We have revised our manuscript again and removed the part about SpinalFlow.
> > >
> > > In our initial response, we discuss SpinalFlow in order to compare the neuromorphic computing based architectures with ANN accelerator based architectures, highlighting their differences and common starting points. SpinalFlow uses an input buffer for the input spikes corresponding to a postsynaptic neuron and processes the input spikes sequentially. As the sparsity of the spike train increases, the number of spikes to be processed in the queue decreases, thus exploiting the sparsity of the spike train. Similarly, as the connections in the network are pruned, the number of connections to a postsynaptic neuron decreases. Therefore, the number of input spikes to a postsynaptic neuron decreases, which has a similar effect to increasing the sparsity of the spike train. However, due to differences in encoding and the fact that the original SpinalFlow may not directly support unstructured neurons, we recognize that the experimental results in the original paper may not prove our point. Therefore, we have revised our manuscript and removed the discussion of SpinalFlow.
> > >
> > > We sincerely appreciate your effort in reviewing our work. If you still have any further comments or questions, please let us know and we will be glad to address your further concerns.

---

> ### Author Response · Authors · 2023-11-21
> **Thanks for your time and hope our responses helpful for your re-assessment of our work**
>
> Dear reviewer BChF, we hope that our response has adequately addressed your concerns regarding our paper and serves as a reference for your reassessment of our work. If you have any further comments or questions, please do not hesitate to let us know. We would be glad to provide a follow-up response. Thank you for your time and effort in reviewing our work.

---

### Official Review · Reviewer_iYS5 · 2023-11-01

**Soundness:** 3 good
**Presentation:** 3 good
**Contribution:** 3 good
**Rating:** 8
**Confidence:** 5

**Summary:**

This paper is dedicated to improving the energy efficiency of SNNs. It defines the energy consumption model and explores a new route to optimizes energy efficiency directly. The authors propose a fine-gained pruning framework that combines unstructured weight and neuron pruning, along with a novel energy penalty term to address the ill-posed problem of jointly pruning neurons and weights. The paper demonstrates the effectiveness of the proposed methods on various datasets and shows that they outperform existing state-of-the-art methods in reducing energy consumption while maintaining comparable performance.

**Strengths:**

1.This paper presents a pioneering effort in directly optimizing the energy efficiency of SNNs, offering novel ideas for reducing energy consumption in neuromorphic computing.

2.The conclusions are impressive. The authors highlight the fact that having fewer weight (parameters) does not necessarily translate to lower energy consumption, which prompts us to reevaluate the efficacy of pruning methods in SNNs. Many existing pruning works may not effectively reduce energy consumption.

3.The structure is clear and easy to follow.

4.The experimental results show that the proposed method can achieve the state-of-the-art balance between accuracy and energy efficiency.

**Weaknesses:**

1.Some portions of the paper lacks clarity. For example, Figure 1 is not sufficiently elucidated. The authors illustrate in the motivation section that Figure 1 shows how fine-grained pruning significantly reduces synaptic connections, but it is challenging to grasp. Furthermore, some variables are not clearly defined, such as $d_n$ and $d_w$ in Equation (6).

2.There are some grammar errors that require improvement.
Page 4: 'better trade-off' should be 'a better trade-off',
Page 7: 'refers to' should be ' refer to',
Page 9: 'outperformes' should be 'outperforms'.

**Questions:**

1. For Figure 1, additional details regarding the experimental settings and result analysis should be added in the appendix.

2.Why there is not a element-wise multiplication for $m_n$ in Equation (6)? Why it is different from the masks of weights $m_w$?

3.Before Equation (6), it appears that the order may be reversed. Specifically, $m_w$ should be the masks of weights and $m_n$ should be the masks of neurons.

---

> ### Author Response · Authors · 2023-11-15
> **Rebuttal by Authors**
>
> We sincerely appreciate your approval of our work and helpful feedback. We will incorporate all your suggestions and answer your questions.
>
> > Figure 1 is not sufficiently elucidated.
>
> Thanks for pointing it out. The purpose of Figure 1 is to provide readers with a visual representation of fine-grained pruning. The left figure shows neurons, weights, and synaptic connections in a convolutional layer with $5\times 5$ input and a $3\times 3$ filter.  In order to avoid visual clutter caused by an excessive number of synaptic connections, we have chosen to display only 9 synaptic connections that correspond to one of the postsynaptic neurons. The right figure shows the effect of different pruning techniques on the number of synaptic connections. Each subfigure in the right figure is labeled with the number of synaptic connections that remain after pruning. Similarly, due to the excessive number of synaptic connections, we have not visualized the synaptic connections, which seems to have caused confusion. We have added these notes in the revised version.
>
>
> > Why there is not an element-wise multiplication for $m_n$ in Equation (6)? Why it is different from the masks of weights $m_w$?
>
> This is because neurons are not considered as learnable parameters within the network, and as a result, they do not appear in Equation 6. Therefore, we only include $\boldsymbol m_n$ to denote the mask of neurons in Equation 6.
>
>
> > some variables are not clearly defined, such as $d_n$ and $d_w$ in Equation (6).
>
> Thanks for your comment. In Equation (6), $d_n$ and $d_w$ denote the number of neurons and weights in the network $f$, respectively. We have added these notes in the revised version.
>
> > There are some grammar errors that require improvement.
> > Before Equation (6), it appears that the order may be reversed.
>
> Thank you for pointing out these errors. We have fixed them in the revised version.

---

> > ### Comment · Reviewer_iYS5 · 2023-11-22
> > **Thanks for the response.**
> >
> > I appreciate your time to consider my suggestions and improve the final version of the paper. The authors have presented a nice work to optimize the energy efficiency of SNNs for neuromorphic computing. After reading all reviews, I still believe this paper is deserving of acceptance.

---

### Official Review · Reviewer_gonx · 2023-11-01

**Soundness:** 3 good
**Presentation:** 3 good
**Contribution:** 2 fair
**Rating:** 6
**Confidence:** 4

**Summary:**

The paper proposes a framework towards unstructured pruning of weights and neurons in Spiking Neural Networks for high sparsity and significant reduction in spikes. Masks are learned during training to trade off accuracy loss and energy consumption. Experiments are provided to demonstrate competitive reduction in spiking operations with higher accuracy than previous works.

**Strengths:**

The authors provide a sound approach towards learned pruning for energy at high-granularity. Experimental details including hyperparameter sensitivity and ablation are addressed and elaborated in sufficient detail. From their results, the proposed method is demonstrated to achieve a competitive accuracy-energy trade-off compared to recent works. Lastly, the paper is clearly written with enough details provided for reproducibility.

**Weaknesses:**

1.The central idea doesn't appear to offer a breakthrough. The hyperparameter search may not sufficiently differentiate this work. Although combining energy factors into training is a novel approach for SNNs, it has previously been utilized in other contexts, such as artificial neural networks, for various objectives. Such as in the following paper:

Salehinejad, H., & Valaee, S. (2021). Edropout: Energy-based dropout and pruning of deep neural networks. IEEE Transactions on Neural Networks and Learning Systems, 33(10), 5279-5292.

2. The comparison with the state-of-the-art, especially concerning the ImageNet dataset, lacks comprehensiveness. Only one previous work is considered.

3. While the proxy measure of energy-efficiency using synaptic operations (SOPs) is justified from prior neuromorphic hardware works, the paper could improve its claim for energy efficiency by providing actual energy saved in implementation on neuromorphic hardware.

4. In paragraph 3 of Section 5.2, the proposed energy model is claimed to be validated by energy estimates of GradR and STDS. However, as the energy measure used (SOPs) is itself defined by the energy model, this claim seems circular.

**Questions:**

In Table 2 (comparison with state-of-the-art), the accuracy for GradR-CIFAR10 and STDS-ImageNet are obtained from the original papers, but the percentage of connections reported differs from the original papers. Please clarify these differences (i.e. if there is a difference in definition).

---

> ### Author Response · Authors · 2023-11-15
> **Rebuttal by Authors Part (1/3)**
>
> We sincerely appreciate your thorough review of our paper and the invaluable insights you provided. We are grateful for the opportunity to address the points you raised.
>
>
> > Although combining energy factors into training is a novel approach for SNNs, it has previously been utilized in other contexts, such as artificial neural networks, for various objectives. Such as in the following paper [1].
>
> We would like to clarify that the concept of "energy" in the energy consumption model proposed in our paper and the "energy" referred to in [1] are distinct and unrelated. In our paper, the energy consumption model is a model for estimating the energy consumption of SNNs, with "energy" specifically referring to the energy consumption of the network itself. On the other hand, the Energy-Based Model (EBM) mentioned in [1] is a type of machine learning model where "energy" represents a mathematical function that measures the compatibility between input data and the model's parameters, without any connection to energy consumption. Therefore, it is important to note that our proposed method and [1] are entirely different approaches with distinct goals. Our method mainly focuses on reducing the energy consumption of SNNs, while the approach in [1] is designed to reduce the model parameters.
>
>
> > The comparison with the state-of-the-art, especially concerning the ImageNet dataset, lacks comprehensiveness. Only one previous work is considered.
>
>
> Thanks for pointing it out! Actually, there is limited existing work on SNN pruning, and currently, only STDS [2], the state-of-the-art SNN pruning method, provides experimental results on the ImageNet dataset. As a result, we primarily focus on comparing our method with STDS. It is worth noting that the original paper of STDS mentions the results of other pruning methods (GradR [3] and ADMM [4]) on the ImageNet dataset, which were implemented by the authors themselves. However, the performance of these methods significantly lags behind STDS, and no relevant code implementations are provided. Consequently, we have not included these methods in our comparison.
>
> To provide a more comprehensive evaluation, we are working on reproducing the results of these pruning methods (GadR and ADMM) by our implementation. Unlike the comparison in STDS, our comparison requires additional metrics such as the number of SOPs and connections, in addition to the accuracy and number of parameters already provided. Therefore it is necessary to reproduce these experiments and obtain the trained models to measure these metrics, which is a time-consuming process. We have reproduced the experiments of ADMM and have presented the results in Table R1. The experimental results indicate that compared to both STDS and ADMM, our method achieves comparable performance while significantly reducing energy consumption.
>
> We have added the comparison in the appendix. The GradR experiments are still in progress and we will report the results as soon as they are available.
>
> **Table R1. Comparison on ImageNet dataset**
>
> | Method   | Arch.        | T    | Top-1 Acc. (%) | Acc. Loss (%) | Avg. SOPs (M) | Ratio     | Conn. (%) | Param (M) |
> | -------- | ------------ | ---- | -------------- | ------------- | ------------- | --------- | --------- | --------- |
> | ADMM     | SEW ResNet18 | 4    | 59.48          | -3.74         | 402.75        | 3.33      | 29.25     | 1.94      |
> | ADMM     | SEW ResNet18 | 4    | 55.85          | -7.37         | 262.39        | 5.11      | 21.04     | 1.24      |
> | ADMM     | SEW ResNet18 | 4    | 50.57          | -12.65        | 173.10        | 7.76      | 15.68     | 0.75      |
> | STDS     | SEW ResNet18 | 4    | 62.84          | -0.34         | 1082.60       | 1.24      | 77.54     | 7.42      |
> | STDS     | SEW ResNet18 | 4    | 61.51          | -1.67         | 512.95        | 2.61      | 34.79     | 2.38      |
> | STDS     | SEW ResNet18 | 4    | 61.30          | -1.88         | 424.07        | 3.15      | 29.29     | 1.94      |
> | STDS     | SEW ResNet18 | 4    | 59.93          | -3.25         | 273.39        | 4.89      | 21.08     | 1.24      |
> | STDS     | SEW ResNet18 | 4    | 58.06          | -5.12         | 182.49        | 7.33      | 15.71     | 0.75      |
> | STDS     | SEW ResNet18 | 4    | 56.28          | -6.90         | 134.64        | 9.94      | 12.95     | 0.52      |
> | **Ours** | SEW ResNet18 | 4    | **61.89**      | **-1.29**     | **311.70**    | **4.29**  | **28.60** | 3.93      |
> | **Ours** | SEW ResNet18 | 4    | **60.00**      | **-3.18**     | **177.99**    | **7.52**  | **19.31** | 3.10      |
> | **Ours** | SEW ResNet18 | 4    | **58.99**      | **-4.19**     | **116.88**    | **11.45** | **14.19** | 2.62      |

---

> > ### Author Response · Authors · 2023-11-15
> > **Rebuttal by Authors Part (2/3)**
> >
> > > While the proxy measure of energy-efficiency using synaptic operations (SOPs) is justified from prior neuromorphic hardware works, the paper could improve its claim for energy efficiency by providing actual energy saved in implementation on neuromorphic hardware.
> >
> > Thanks for your constructive suggestion! Unfortunately, we currently do not have access to neuromorphic hardware that would allow us to deploy the sparse models generated by our method and measure their actual energy consumption. Thus, we use SOPs as a proxy metric for energy consumption, which aligns with previous works such as ESLSNN [5] mentioned in the main text. Additionally, we would like to highlight that the majority of research papers on SNNs compare energy efficiency using SOPs as the metric [6-8]. This established practice allows for consistent and meaningful comparisons across different studies in the field.
> >
> > > In paragraph 3 of Section 5.2, the proposed energy model is claimed to be validated by energy estimates of GradR and STDS. However, as the energy measure used (SOPs) is itself defined by the energy model, this claim seems circular.
> >
> > We apologize for any confusion caused by our previous statements. We would like to clarify our intended meaning. In our analysis, we have found that the number of SOPs is proportional to the number of connections. Our energy consumption model, as described in Equation 1, aims to refine the SOPs into two components: the number of spikes and the number of connections. Assuming that the firing rates of neurons remain constant, we expect the SOPs to be proportional to the number of connections. The results we have obtained (41.89M SOPs/2.15% connections vs. 26.81M SOPs/1.35% connections) confirm our assumption and provide validation for our objective of reducing energy consumption by decreasing the number of connections. We have altered the expression in the revised version.
> >
> > >In Table 2 (comparison with state-of-the-art), the accuracy for GradR-CIFAR10 and STDS-ImageNet are obtained from the original papers, but the percentage of connections reported differs from the original papers. Please clarify these differences (i.e. if there is a difference in definition).
> >
> >
> > Yes, there is a difference in the definition of "connections" in our paper compared to GradR and STDS. In our paper, "connections" specifically refers to synaptic connections, as described in Section 3.1 of the Energy Consumption Model. On the other hand, in GradR and STDS, "connections" actually refers to the weights of the model. The "Connection (%)" data in the original paper of GradR corresponds to the "Param." in Table 2, which is calculated by multiplying the "Connection (%)" obtained from the original paper with the number of parameters of the dense model.
> >
> > We use a different definition of "connection" in order to distinguish the synaptic connections and their corresponding weights. It is important to note that weights can be reused in multiple connections, whereas synaptic connections represent the actual connections between neurons. For instance, in the convolution layer illustrated in Figure 1 of the main text, there are 169 synaptic connections, while the weights are only 9.
> >
> > To clarify this distinction, we have provided an explanation in the footnotes of Table 2 in the revised version of our paper.

---

> > > ### Author Response · Authors · 2023-11-15
> > > **Rebuttal by Authors Part (3/3)**
> > >
> > > [1] Salehinejad, H., & Valaee, S. (2021). Edropout: Energy-based dropout and pruning of deep neural networks. *IEEE Transactions on Neural Networks and Learning Systems*, 33(10), 5279-5292.
> > >
> > > [2] Chen, Y., Yu, Z., Fang, W., Ma, Z., Huang, T., & Tian, Y. (2022). State Transition of Dendritic Spines Improves Learning of Sparse Spiking Neural Networks. *International Conference on Machine Learning*, 1-9.
> > >
> > > [3] Chen, Y., Yu, Z., Fang, W., Huang, T., & Tian, Y. (2021). Pruning of deep spiking neural networks through gradient rewiring. *Proceedings of the International Joint Conference on Artificial Intelligence*, 1713–1721.
> > >
> > > [4] Deng, L., Wu, Y., Hu, Y., Liang, L., Li, G., Hu, X., Ding, Y., Li, P., & Xie, Y. (2021). Comprehensive SNN compression using ADMM optimization and activity regularization. *IEEE Transactions on Neural Networks and Learning Systems*, 2791-2805.
> > >
> > > [5] Shen, J., Xu, Q., Liu, J. K., Wang, Y., Pan, G., & Tang, H. (2023). ESL-SNNs: An Evolutionary Structure Learning Strategy For Spiking Neural Networks. *Proceedings of the AAAI Conference on Artificial Intelligence*, 1-8.
> > >
> > > [6] Zheng, H., Wu, Y., Deng, L., Hu, Y., & Li, G. (2021). Going deeper with directly-trained larger spiking neural networks. *Proceedings of the AAAI Conference on Artificial Intelligence*, 1-12.
> > >
> > > [7] Fang, W., Yu, Z., Chen, Y., Huang, T., Masquelier, T., & Tian, Y. (2021). Deep residual learning in spiking neural networks. *Advances in Neural Information Processing Systems*, 34, 21056-21069.
> > >
> > > [8] Zhou, Z., Zhu, Y., He, C., Wang, Y., Yan, S., Tian, Y., & Yuan, L. (2023). Spikformer: When spiking neural network meets transformer. *Proceedings of the International Conference on Learning Representations*, 1-17.

---

> ### Author Response · Authors · 2023-11-21
> **Thanks for your time and we hope our response helps to address your questions**
>
> Dear reviewer gonx, we sincerely hope that our response to your review will assist in addressing your concerns and serve as a reference for your reassessment of our work. If you have any further comments or questions, please do not hesitate to let us know. We will be glad to provide a follow-up response. Thank you for your time and effort in reviewing our work.

---

> ### Author Response · Authors · 2023-11-22
> **Thank you for your time and look forward to your feedback**
>
> Dear Reviewer gonx,
>
> We notice that your initial review can be grouped into an overall concern and four detailed concerns.
>
> 1. The main concern is about the central idea of our paper. We have clarified that the concept of "energy" in the energy consumption model proposed in our paper and the "energy" referred to in [1] are distinct and unrelated in our response. Our proposed method and [1] are entirely different approaches with distinct goals.
> 2. The secondary concern is the comparison with the state-of-the-art methods. We have added more comparisons with ADMM on ImageNet dataset in our response and the revised version.
> 3. The third concern is about the implementation on neuromorphic hardware. Unfortunately, we currently do not have access to neuromorphic hardware that would allow us to deploy the sparse models to measure their actual energy consumption. Thus, we use SOPs as a proxy metric for energy consumption, which aligns with previous works. Additionally, we would like to highlight that the majority of research papers on SNNs compare energy efficiency using SOPs as the metric.
> 4. The fourth concern is about the unclear statements. We have clarified our intended meaning in our response and altered the expression in the revised version.
> 5. The last question is about the difference in definition compared with previous works. We have explained the distinction and why we use a different definition in our response.
>
> Based on these facts and positive feedback from other reviewers, we sincerely hope you can reconsider your initial rating. If you still have any further comments or questions, please let us know and we will be glad to address your further concerns.

---

> > ### Comment · Reviewer_gonx · 2023-11-22
> >
> > Thanks for the clarification, [1] is just used to show that it is not very novel to use machine learning optimizer to train the model and concurrently pursue one “system level” metrics for any objective. The [1] I mentioned is just one of the objectives. In fact, there are papers to optimise the “energy consumption” of models on hardware as the objective. For example, Z. Wang, T. Luo, R. S. M. Goh, W. Zhang and W. -F. Wong, "Optimizing for In-Memory Deep Learning With Emerging Memory Technology," in IEEE Transactions on Neural Networks and Learning Systems. It is better to discuss and differentiate your work from such related work. In addition to that, it is good to see you provide results on ImageNet. Therefore, I raised my rating.

---

### Meta-Review · Area_Chair_xMM9 · 2023-12-13

**Metareview:**

This paper proposes unstructured pruning for increasing sparsity and hence decreasing the energy consumption in spiking neural networks. Reviewers agree that this is the first work to address this problem.

Reviewers asked for more comparisons and authors provided some. Authors rebuttals carefully addressed all the concerns of the reviewers and 2 of the reviewers raised their scores.

Reviewer BChF gave a score of 3 and the only reason for that is that the method is not applicable to all accelerator architectures. I do not think that it is a good reason to reject this work. It is ok if some solutions are specific to some accelerators.

Overall, this is a solid work that is worth publishing!

**Justification For Why Not Higher Score:**

This work is not impressive enough to be an oral.

**Justification For Why Not Lower Score:**

I think this work deserves a spotlight because there is not a lot of effort into spiking neural networks and it is good to highlight this to encourage more people to work on SNNs and also to provide diversity in the topics for the spotlight.

---

### Decision · Program_Chairs · 2024-01-16

Accept (spotlight)